# Optimization and Generalizability: New Benchmarking for Stochastic Algorithms

## Abstract

The rich deep learning optimization literature reflects our fragmented understanding of what makes a good optimizer and, more importantly, whether improved optimization performance confers higher generalizability. Current literature neglects an important innate characteristic of SGD and variants, their stochasticity, failing to properly benchmark these algorithms and so reveal their performance in the statistical sense. We fill this gap in this paper. Unlike existing work which evaluates the end point of one navigation/optimization trajectory, we utilize and sample from the ensemble of several optimization trajectories, so that we can estimate the stationary distribution of a stochastic optimizer. We cast a wide net and include SGD and noise-enabled variants, flat-minima optimizers, as well as new algorithms we debut in this paper by recasting noise-enabled optimizers under the Basin Hopping framework. Our evaluation considers both synthetic functions with known global and local minima of varying flatness and real-world problems in computer vision and natural language processing. Our benchmarking accounts for the statistical setting, comparing populations of models and testing for statistical significance. Our paper reveals several findings on the relationship between training loss and hold-out accuracy, the comparable performance of SGD, noise-enabled variants, and novel optimizers based on the BH framework; indeed, these algorithms match the performance of flat-minima optimizers like SAM with half the gradient evaluations. We hope that this work will support further research that accounts for the stochasticity of optimizers for deep learning.

## 1 Introduction

While we now frame the training process during deep learning as the optimization of a typically complex, nonconvex objective/loss function, we do not quite understand, nor can we guarantee, what happens during training (Poggio et al., 2020). We rely on gradient-descent (GD) algorithms originally developed and well characterized for convex optimization. Certainly, stochastic gradient descent (SGD), a variant of the GD algorithm for deep learning, has become the cornerstone optimization algorithm for training (Bottou et al., 2018), and its empirical good performance has been reported in many papers across application settings.

Growing theoretical work is attempting to understand when and why SGD and its variants work well or not. The focus is often on the ability of these optimization algorithms, to which one refers as optimizers, to match their performance on the training data on the testing data; that is, the focus is on generalization (Chatterjee, 2020). The body of literature is rich and often reports contradictory findings, but an increasingly popular line of work has been to connect flat, low-loss regions of the landscape with good generalization (Keskar et al., 2016; Baldassi et al., 2020; Foret et al., 2021; Baldassi et al., 2021; Zhao et al., 2022) and then to devise optimizers that bias their exploration of high-dimensional, nonconvex loss landscapes to flat local minima (Izmailov et al., 2018; Foret et al., 2021). Note that since all these algorithms are stochastic (whether through the mini batches or deliberate noise injection), no guarantee can be made that they reach the global minimum.

The rich deep learning optimization literature reflects our fragmented understanding of what makes a good optimizer and, more importantly, whether improved optimization performance confers higher generalizability. This latter point is indeed important to understand, but what we observe uniformly across literature is that findings are reported on *one single* model. Typically, this is the model onto

which an optimizer has converged or is the lowest-loss model from a window of convergence. This practice neglects a fundamental innate characteristic of SGD and its variants (including flat-minima optimizers), their inherent stochasticity. This paper fills this gap and accounts for the stochasticity of deep learning optimizers in the presence of complex, nonconvex loss functions typically associated with real-world tasks for deep learning. In particular, this paper makes the following contributions:

**1. Expanding the current characterization from a single model to a population of models:** A key insight that informs our work in this paper is that a gradient-guided/biased exploration of the loss landscape by an optimizer during training is limited to one trajectory that the optimizer "launches" over the landscape. In the presence of a complex, nonconvex loss landscape, one trajectory affords a local view of the landscape. To better characterize optimizers and remove potential artifacts or biases due to initial/start conditions, we advocate sampling models over several optimization trajectories so as to obtain a nonlocal view of the landscape by an optimizer.

**2. Rigorous comparison over synthetic loss landscapes, real-world tasks, and model architectures:** We rigorously compare optimizers on both synthetic functions with known global and local minima of varying flatness and on real-world problems. We debut new comparison approaches to characterize and compare populations of models and, in particular, introduce statistical significance testing to support any conclusions made from comparisons.

**3. Novel stochastic optimization algorithms under the Basin Hopping (BH) framework:** We include in our systematic comparison not only SGD, two representative noise-enabled variants, and a recent representative of flat-minima optimizers, but also novel noise-enabled optimizers designed as different instantiations of the BH framework for deep learning.

**4. Generalization performance over rate of convergence:** Unlike most literature on optimization for deep learning, we consider generalization performance rather than simply rate of convergence. We do so over a population of models obtained by an optimizer over several optimization trajectories rather than a single model often obtained as representative of the performance of an optimizer. We compare such a population for its generalization performance (to what we refer as SetA later on in the paper) to a population of models that are sampled by the optimizer and that an oracle has determined have the best generalization performance (to what we refer as SetB later on in the paper). Through this setup we test whether optimization performance is a good proxy of generalization performance utilizing hypothesis testing over populations of models.

**5. New benchmarking for stochastic optimizers:** By properly accounting for the stochastic nature of optimizers, we introduce new benchmarking practices and support a growing body of work to understand the relationship between better optimizers and better generalizability, as well as properly characterize the advantages of novel optimizers in the presence of complex, nonconvex loss functions. To support future work, we open-source the code for all algorithms, hyperparameters, and all comparison approaches.

## 2 BACKGROUND AND RELATED WORK

**Stochastic Gradient Descent**   Consider a multi-dimensional variable/parameter space $\mathbf{w} \in \mathbb{R}^p$ and a loss function $f(\mathbf{w})$ that lifts the variable space into a landscape. At every iteration $t \in [T]$, where $T$ is a fixed budget, GD takes a discrete step in the direction of steepest descent and updates $\mathbf{w}_t = \mathbf{w}_{t-1} - \eta \cdot \hat{\mathbf{g}}$, where $\hat{\mathbf{g}}$ is the normalized gradient vector $\mathbf{g} = \nabla f(\mathbf{w}_{t-1})$ of $f$ evaluated at $\mathbf{w}_{t-1}$; $\mathbf{w}_0$ (initial conditions) are sampled at random over the variable space. The "stochastic" in SGD is a key difference from GD and refers to the stochasticity of minibatches (Le et al., 2011; Duchi et al., 2011; Zeiler, 2012; Kingma & Ba, 2015); SGD minimizes the empirical loss $\frac{1}{|\mathcal{B}_t|} \sum_{i \in \mathcal{B}_t} f_i(\mathbf{w}_t)$, where $f_i$ is the loss for a data point $i$ in the minibatch $\mathcal{B}_t$ drawn from the training dataset at iteration $t$ of training. The minibatch construct avoids storing all data in memory and extends SGD to online settings (Bottou et al., 2018; Shalev-Shwartz, 2012; Le et al., 2011).

**Exploration versus Exploitation**   The step size $\eta$ determines how much to "walk" in the direction of the (negative) gradient; a large value risks overshooting and increasingly deviating away from $f$; a small value, while tracking $f$ more faithfully, risks premature convergence to a nearby minimum, possibly missing better ones. Work in (Baydin et al., 2018) proposes optimizing $\eta$ via GD, and recent work automatically extends it to SGD variants (Chandra et al., 2022). However, for non-convex

optimization, the ruggedness/multi-modality of the loss landscape (for which we have increasing evidence (Li et al., 2018; Bosman et al., 2020; Liu et al., 2020)) challenges controlling the balance between exploration (of the entirety of the landscape) and exploitation (of minima) through $\eta$ alone. The initial conditions $\mathbf{w}_0$ can also unduly impact the exploration.

## NOISE-ENABLED VARIANTS OF SGD

Due to GD convergence to stationary points other than local minima (such as saddle points), early work proposed to incorporate randomness in the process, by injecting noise in the gradient (Ge et al., 2015) or the model (Jin et al., 2017). Consider a noise vector $\eta$ drawn at random from $B_0(\rho)$ (a ball centered at the origin with radius $\rho$). In (Ge et al., 2015), this noise is added to the gradient prior to updating the model parameters, as shown in Algorithm 1. Work in (Jin et al., 2017) instead injects noise to the model parameters $\mathbf{w}$ directly, as shown in Algorithm 2, and conditionally, only after a certain number of iterations $\tau$ have been reached AND the magnitude of the gradient has become small. The first condition ensures that time is provided for exploitation via GD. The second condition identifies when a stationary point is reached. We rename these algorithms as NoiseInGradient-GD and NoiseInModel-GD and abbreviate them in the interest of space as NiG-GD and NiG-SGD. Note that the presentation here is for GD, but the SGD variants operate over the minibatches.

---

**Algorithm 1: NiG-GD (Ge et al., 2015)**

1: **Input:** $f(\mathbf{w}), T > 0, \mathbf{w}, \eta, \rho$
2: **Output:** $\mathbf{w}$
3: **while** $t \leq T$ **do**
4:   $\mathbf{g} \leftarrow \nabla f(\mathbf{w})$
5:   $\zeta \in B_0(\rho)$          ▷sample noise
6:   $\mathbf{g} \leftarrow \mathbf{g} + \zeta$          ▷add to gradient
7:   $\mathbf{w} \leftarrow \mathbf{w} - \eta \cdot \mathbf{g}$
8: **end while**

---

**Algorithm 2: NiM-GD (Jin et al., 2017)**

1: **Input:** $f(\mathbf{w}), T > 0, \epsilon \cong 0, \tau > 0, \rho$
2: **Output:** $\mathbf{w}$
3: **while** $t \leq T$ **do**
4:   $\mathbf{g} \leftarrow \nabla f(\mathbf{w}_t)$
5:   **if** $\|\mathbf{g}\| < \epsilon$ and $t > \tau$ **then**
6:     $\zeta \in B_0(\rho)$          ▷sample noise
7:     $\mathbf{w}_t \leftarrow \mathbf{w}_t + \zeta$          ▷add to model
8:     $\mathbf{g} \leftarrow \nabla f(\mathbf{w}_t)$
9:   **end if**
10:   $\mathbf{w} \leftarrow \mathbf{w} - \eta \cdot \mathbf{g}$
11: **end while**

---

Work in (Zhou et al., 2019), though limited to a simple two-layer convolutional neural network (CNN) model, shows that adding annealing noise to the gradient allows SGD to provably converge to a global optimum in polynomial time with arbitrary initialization. Work in (Orvieto et al., 2022) connects injecting noise within GD with smoothing and regularization and shows that independent layer-wise perturbations circumvent the exploding variance term in over-parameterized models, yielding explicit regularization and better generalization. The stated motivation of noise-enabled optimizers is to escape saddle points. There is a rich history and literature on noisy gradient methods based on the Langevin dynamics (LD) (Kennedy, 1990; Neill, 2011; Welling & Teh, 2011; Chaudhari et al., 2017; Ma et al., 2018; Chourasia et al., 2021). Recent work (Banerjee et al., 2022) additionally relaxes the Gaussian noise assumption within the LD framework. In this paper, we focus on the simplest noise-enabled variants of SGD, hoping to extend to LD-based ones in future work. For noise-enabled optimizers, we posit that it is useful to think of them as attempts to increase the exploration capability in a framework of exploration versus exploitation (as is common in stochastic optimization). While following a gradient increases exploitation, adding a perturbation to this via injecting noise in the gradient or directly the model enhances exploration.

## FLAT-MINIMA OPTIMIZERS

Research on the benefit of flat minima (with flatness loosely referring to the curvature of the neighborhood around a local minimum) is contradictory. One could summarize it as follows: Poorly generalizable local minima are sharp (Keskar et al., 2016). SGD has an inherent bias to converge to flat local minima (Smith & Le, 2018). Generalization can improve with further bias towards flat minima (Izmailov et al., 2018; Foret et al., 2021). Sharp minima can generalize for deep nets (Dinh et al., 2017) on a variety of tasks (Kaddour

---

**Algorithm 3: SAM (Foret et al., 2021)**

1: **Input:** $f(\mathbf{w}), T > 0, \mathbf{w}, \eta, \rho > 0$
2: **Output:** $\mathbf{w}$
3: **while** $t \leq T$ **do**
4:   $\mathbf{g} \leftarrow \nabla f(\mathbf{w})$
5:   $\hat{\mathbf{g}} \leftarrow \frac{\mathbf{g}}{\|\mathbf{g}\|}$          ▷normalize gradient
6:   $\zeta \leftarrow \rho \cdot \hat{\mathbf{g}}$          ▷get perturbed vector
7:   $\mathbf{w} \leftarrow \mathbf{w} + \zeta$          ▷modify model
8:   $\mathbf{g} \leftarrow \nabla f(\mathbf{w})$          ▷update gradient
9:   $\mathbf{w} \leftarrow \mathbf{w} - \eta \cdot \mathbf{g}$
10: **end while**

---

et al., 2022). Nonetheless, researchers seek novel optimization algorithms biased in some manner towards flat local minima. We single out here as representative the Sharpness Aware Minimization (SAM) algorithm (Foret et al., 2021). SAM minimizes the maximum loss around a neighborhood of the current SGD iterate but requires an additional forward/backward pass for each parameter update. As shown in Algorithm 3, rather than sampling a noise vector in $B_0(\rho)$, a deterministic vector $\zeta$ (of magnitude $\rho$) in the direction of the gradient is added to the model parameters; There is no true noise injection, as $\rho$ is an input parameter. The gradient is calculated twice (lines 4 and 8). SAM occupies its own category, given that it does not inject any noise but through a deterministic vector aims to get out of a stationary point. Attempts have been made to understand SAM. Work in (Bartlett et al., 2022) provides bounds on SAM's rate of convergence and shows that, when applied with a convex quadratic objective, for most random initializations, SAM converges to a cycle that oscillates between either side of the minimum in the direction with the largest curvature. Comparison of SAM to Stochastic Weight Averaging (SWA) (Izmailov et al., 2018) on diverse tasks (vision, NLP, etc.) shows no clear winner on convergence to flat minima, SAM converging to non-flat minima, and non-flat minima sometimes having better generalization (Kaddour et al., 2022).

## 3 BENCHMARKING SETUP

We consider both synthetic functions that allow us to characterize the optimization dynamics of an optimizer in a controlled setting and real-world tasks where we do not know the loss landscape.

### OPTIMIZATION DYNAMICS IN A CONTROLLED SETTING

It is informative to characterize the optimization dynamics of an optimizer in a controlled setting; synthetic nonconvex functions with known minima provide us with that. We have compiled several synthetic functions, three of which are visualized in Figure 1 (with more in the Supplementary Material). The synthetic functions we have selected are rich in global and local minima of varying sharpness and they span the spectrum of structured to unstructured (e.g., having broad plateaus or numerous local minima). To capture the behavior of an optimizer over a synthetic nonconvex function, we sample the "stationary distribution" of an optimizer (end-points of its converged optimization trajectories) by "restarting" the optimizer $R$ times. These $R$ times are also known as random restarts in optimization literature. In each restart, we sample $\mathbf{w}_0$ non-uniformly at random; From each initial condition, for each optimizer, the resulting trajectory (of consecutive models, which we typically visualize during training via training loss) is continued for a fixed budget of $I$ iterations. The end-model of each trajectory is added to a population of models. This population is analyzed for its coverage of the various known global and local minima of a synthetic landscape, affording us a global view of any innate biases of an optimizer towards particular minima in a synthetic landscape.

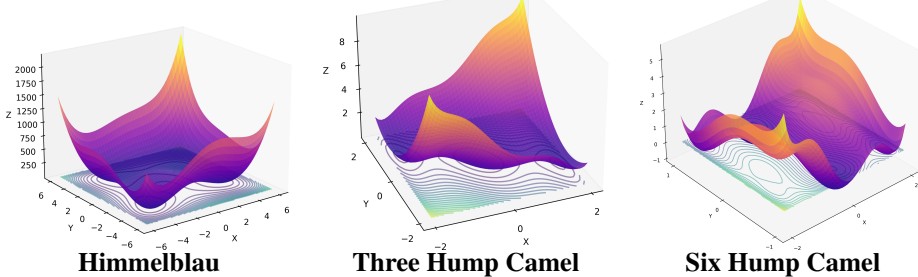

| Himmelblau | Three Hump Camel | Six Hump Camel |

Figure 1: Himmelblau: $f(x, y) = (x^2 + y - 11)^2 + (x + y^2 - 7)^2$. Three-Hump Camel: $f(x, y) = 2 \cdot x^2 - 1.04 \cdot x^4 + \frac{x^6}{6} + xy + y^2$. Six-Hump Camel: $f(x, y) = (4 - 2.1x^2 + \frac{x^4}{3}) \cdot x^2 + xy + (-44 \cdot y^2) \cdot y^2$. The locations of the (global and local) minima of each of these functions are listed in the Supplementary Material.

### STATISTICAL CHARACTERIZATION ON REAL-WORLD PROBLEMS

We also characterize optimizers on the following real-world tasks: CIFAR 10, CIFAR 100 (Krizhevsky et al., 2009), and ImageNet (Deng et al., 2009) image classification problem using ResNet-50 (He et al., 2016), emotion classification on GoEmotions (Demszky et al., 2020) and TweetEval (Barbieri et al., 2020) datasets. We select these tasks to account for both settings of accuracy or macro-F1 as indicators of generalizability. In text mining and NLP tasks, macro-F1 is more

popular due to data imbalance challenges in multi-class classification tasks. In the Supplementary Material we expand the analysis to account for different model architectures for each task.

In real-world tasks we do not know the loss landscape and so cannot make use of the above approach. Instead, to account for the stochastic nature of an optimizer, we propose the following approach. We sample from a given number $Tr$ of random restarts; $Tr < R$ here because of the typically higher cost of an optimizer on real-world loss landscapes versus synthetic ones. The key insight is that we treat each trajectory as a local view of a loss landscape afforded by an optimizer and so sample from a trajectory not just the last/converged model, but $L$ models.

We explore two settings to obtain two distinct populations over models "sampled" by an optimizer over several trajectories: (1) from each trajectory, select the $L$ lowest-loss models; (2) from each trajectory, select the $L$ models with highest generalization capability (accuracy or macro-F1 depending on the task). The two resulting populations (to which we refer respectively as SetA and SetB) are compared via statistical tests to elicit any statistically significant differences and so obtain a global/population view on whether models selected by loss are conducive to high generalization capability. By comparing populations of models, we can also better compare two given optimizers and not risk conclusions based on one arbitrary or hand-selected models. For instance, we can compare via statistical tests SetA obtained by Algorithm X to SetA obtained by Algorithm Y to determine which one is better. Our proxy for "better" is test set accuracy or macro-F1. We consider 10 different optimizers.

## 4 Broadening Stochastic Optimizers under BH Umbrella: New Noise-Enabled Optimizers

The four core algorithms we analyze are SGD, NoiseInModel-GD/SGD (which we abbreviate as NiM-GD/SGD from now on), NoiseInGradient-GD/SGD (which we abbreviate as NiG-GD/SGD, and SAM, described in Section 2; the pseudocodes of the latter are presented above. By enabling noise as NiG or NiM, and varying over BH, MonotononicBH, or MetropolisBH, we obtain six more algorithms, to which we refer from now on as NiG-BH, NiM-BH, NiG-MBH and NiM-MBH ('M' for Monotonic), and NiG-MpBH and NiM-MpBH ('Mp' for Metropolis). These algorithms are instantiations of the BH framework for deep learning optimization. The BH framework allows one to incorporate noise in a principled manner. For clarity, we limit our algorithmic exposition to GD, but our evaluation setting considers the minibatch version of the BH algorithms (SGD over GD).

While not presented in this manner, noise-enabled optimizers combine two components, one that exploits the landscape via a local search (the gradient-based model update) and one that explores the landscape via injecting noise in the gradient or the model. These two are core components of the BH framework, which we respectively name `LclSearch` and `Perturb`. The BH framework has a rich history in optimization literature (Olson et al., 2012) and has been adapted for multi-basin landscapes of actuated physical and biological systems (Molloy et al., 2016; Maximova et al., 2015; 2017; 2018). The framework is related in Algorithm 4; as presented, it permits a more general stopping criterion than a fixed budget $T$. BH iterates between minima $Y_i$ in the parameter space, to which `LclSearch` maps a point $X_i$. $X_{i>0}$ are obtained by perturbing the current minimum $Y_i$, and the `Perturbation` component broadens the injection of noise. The distinction between $X$ and $Y$ to present minima clearly, but both correspond to model parameters. Line 9 in Algorithm 4 makes this particular presentation monotonic-BH. Removing the condition in line 9 provides us with the general BH formulation. Variations include not enforcing strict monotonocity but allowing small increases in $f$ with some probability, resulting in a Metropolis versions.

In this paper we instantiate the BH framework for deep learning optimization. The `LclSearch` is the gradient-based model update (the discrete step in the direction of steepest descent). The `Perturbation` component can be implemented in two different ways, either to inject noise in the gradient or the model directly, resulting in two different instantiations, to which we refer as NiG-BH and NiM-BH, respectively. Note that if monotonicity is enforced (as in line 9 in Algorithm 4, then one obtains NiG-MBH (shown in Algorithm 8) and NiM-MBH (shown in Algorithm 9). We note that in our implementation, as shown in Algorithm 5, `LclSearch` carries out $\tau < T$ iterations of gradient descent, or terminates earlier if the gradient flattens. `PerturbModel`, shown in Algorithm 6 is an implementation of `Perturb` by injecting noise (vector $\zeta$) in the model. The returned model parameter is only a candidate (line 7), given the monotonicity constraint (line 8).

Equivalently, injecting noise directly in the gradient can be implemented, shown in PerturbGradient in Algorithm 7.

---

**Algorithm 4: Monotonic BH**

1: **Input:** $f(\mathbf{w})$
2: **Output:** $\mathbf{w}$
3: $i \leftarrow 0$
4: $X_i \leftarrow$ random initial point
5: $Y_i \leftarrow LclSearch(X_i)$
6: **while** NOT STOP **do**
7:     $X_{i+1} \leftarrow Perturb(Y_i)$
8:     $Y_{i+1} \leftarrow LclSearch(X_{i+1})$
9:     **if** $f(Y_{i+1}) < f(Y_i)$ **then**
10:         $i \leftarrow i + 1$
11:     **end if**
12: **end while**

---

**Algorithm 5: LclSearch**

1: **Input:** $f(\mathbf{w}), \mathbf{w}, \tau > 0, \eta, \epsilon$
2: **Output:** $\mathbf{w}, t$
3: **while** $t < \tau$ **do**
4:     $\mathbf{g} \leftarrow \nabla f(\mathbf{w})$
5:     **if** $|\mathbf{g}| < \epsilon$ **then**
6:         Terminate
7:     **end if**
8:     $\mathbf{w} \leftarrow \mathbf{w} - \eta \cdot \mathbf{g}$
9: **end while**

---

**Algorithm 6: PerturbModel**

1: **Input:** $\mathbf{w}, \rho$
2: **Output:** $\mathbf{w}$
3: $\zeta \in B_0(\rho)$
4: $\mathbf{w} \leftarrow \mathbf{w} + \zeta$

---

**Algorithm 7: PerturbGradient**

1: **Input:** $\mathbf{g}, \rho$
2: **Output:** $\mathbf{g}$
3: $\zeta \in B_0(\rho)$
4: $\mathbf{g} \leftarrow \mathbf{g} + \zeta$

---

**Algorithm 8: NiG-BH**

1: **Input:** $f(\mathbf{w}), T > 0, \epsilon \cong 0, \tau > 0, \eta, \rho$
2: **Output:** $\mathbf{w}$
3: $(\mathbf{w}, \Delta t) \leftarrow LclSearch(f, \mathbf{w}, \tau, \eta, \epsilon)$
4: $t \leftarrow t + \Delta t$
5: **while** $t \leq T$ **do**
6:     $\mathbf{g} \leftarrow \nabla f(\mathbf{w}_t)$
7:     $\mathbf{g} \leftarrow PerturbGradient(\mathbf{g}, \rho)$
8:     $\mathbf{w} \leftarrow \mathbf{w} - \eta \cdot g$
9:     $(\mathbf{w}, \Delta t) \leftarrow LclSearch(f, \mathbf{w}, \tau, \eta, \epsilon)$
10:     $t \leftarrow t + \Delta t$
11: **end while**

---

**Algorithm 9: NiM-MBH**

1: **Input:** $f(\mathbf{w}), T > 0, \epsilon \cong 0, \tau > 0, \eta, \rho$
2: **Output:** $\mathbf{w}$
3: $(\mathbf{w}, \Delta t) \leftarrow LclSearch(f, \mathbf{w}, \tau, \eta, \epsilon)$
4: $t \leftarrow t + Deltat$
5: **while** $t \leq T$ **do**
6:     $\mathbf{w} \leftarrow PerturbModel(\mathbf{w}, \rho)$
7:     $(\mathbf{w}_c, \Delta t) \leftarrow LclSearch(f, \mathbf{w}, \tau, \eta, \epsilon)$
8:     $t \leftarrow t + \Delta t$
9:     **if** $f(\mathbf{w}_c) < f(\mathbf{w})$ **then**
10:         $\mathbf{w} \leftarrow \mathbf{w}_c$
11:     **end if**
12: **end while**

---

The BH framework is rich and permits various algorithmic instantiations to assess the exploration-exploitation balance. In this paper we debut and analyze the BH, monotonic BH (MBH), and Metropolis BH (MpBH); the latter replaces the conditional line 9 in Algorithm 4 with the Metropolis criterion (related in pseudocode in the Supplementary Material). In each of these, we investigate adding noise in the gradient or in the model.

## 5 MODEL POPULATION ANALYSIS ON SYNTHETIC NONCONVEX LOSS LANDSCAPES

Table 1 shows the stationary distribution (end points of 500 trajectories, each initiated from a random point sampled uniformly at random over the domain of a function) for each of the 10 algorithms in terms of percentages of converged models over the known minima of the synthetic landscapes. These are the "base" versions of the algorithms with no hyperparameter tuning. For each synthetic function, the global minima are listed first, followed by the local minima. Flatter minima are listed before sharper ones. The Supplementary Material provides visualizations of end-points over synthetic landscapes and adds three more synthetic landscapes to our analysis.

Several observations emerge. First, the Six-Hump Camel function presents more challenging for all optimizers. The population of the first global minimum is low, and the percentage of "non-converged" trajectories is higher (the number of end-points that do not fall in any of the known minima (indicated by 'Else' in the tables). NiG-GD and SAM do particularly poorly on this function, with 38% and 34% of the (population of) models respectively not falling in any of the known

| Algorithms | Himmelblau | | | | | Three-Hump Camel | | | | Six-Hump Camel | | | | | | |
|---|---|---|---|---|---|---|---|---|---|---|---|---|---|---|---|---|
| | GM1 | GM2 | GM3 | GM4 | Else | GM | LM1 | LM2 | Else | GM1 | GM1 | LM1 | LM2 | LM3 | LM4 | Else |
| GD | 28 | 25 | 23 | 24 | 0 | 32 | 35 | 33 | 0 | 4.25 | 23.5 | 22.75 | 17.25 | 1.5 | 10.25 | 20.5 |
| NiG-GD | 28 | 23 | 23 | 25 | 1 | 31 | 35 | 34 | 0 | 13.5 | 16.4 | 14.6 | 15.5 | 0.4 | 1.6 | 38 |
| NiM-GD | 30 | 27 | 23 | 20 | 0 | 34 | 34 | 32 | 0 | 7.375 | 18.38 | 19.88 | 16.5 | 10 | 9.86 | 17 |
| SAM | 30 | 23 | 20 | 20 | 7 | 30 | 27 | 25 | 18 | 11.63 | 16. 1 | 19 | 16.13 | 0.75 | 3 | 34 |
| NiG-BH | 27 | 25 | 25 | 22 | 1 | 32 | 35 | 33 | 0 | 6.25 | 21.75 | 21.75 | 17.75 | 2. | 8.25 | 22 |
| NiM-BH | 27 | 25 | 24 | 24 | 0 | 33 | 35 | 31 | 1 | 6.5 | 20.25 | 21.75 | 17.25 | 2.25 | 10.25 | 22 |
| NiG-MBH | 22 | 27 | 23 | 28 | 0 | 33 | 33 | 34 | 0 | 5.75 | 23.75 | 24.25 | 16.25 | 2 | 11 | 17 |
| NiM-MBH | 30 | 23 | 20 | 20 | 7 | 34 | 32 | 34 | 0 | 5.25 | 24.75 | 24.25 | 16.5 | 1.75 | 11.5 | 12 |
| NiG-MpBH | 25 | 28 | 21 | 25 | 1 | 30 | 35 | 35 | 5 | 5.75 | 23.75 | 24.5 | 17.25 | 1.15 | 11.25 | 16.3 |
| NiM-MpBH | 28 | 25 | 23 | 23 | 1 | 32 | 36 | 32 | 0 | 6 | 22 | 22.75 | 16 | 2 | 9 | 22 |

Table 1: The stationary distribution (reported in % for each entry) for the Himmelblau, Three-Hump Camel, and Six-Hump Camel function for each algorithm. The locations of the global minima (GM) and local minima (LM) for each function are listed in the Supplementary Material.

minima. However, the stationary distribution of these two optimizers is skewed away from LM3 and LM4, which are the sharpest local minima (note that minima are ordered by sharpness, from low to high, in our exposition). Without any particular considerations, just noise in the gradient, NiG-GD matches SAM's performance in skewing the stationary distribution towards the global and flatter minima. This skewness is not observed in the other optimizers, as expected. It is interesting that the BH optimizers have more end-points converging over minima than other regions of the landscape. In the rest of the analysis, we exclude algorithms based on monotonic BH. Their greedy nature, while exploiting well certain synthetic landscapes, makes them prone to premature convergence on complex, real-world landscapes (data not shown), a behavior that is well-documented in complex optimization problems (Olson et al., 2012).

# 6 MODEL POPULATION ANALYSIS ON REAL-WORLD TASKS

As related earlier, we obtain a population of models that are "samples" of the view obtained by a particular optimizer of a loss landscape. We set $Tr = 5$ and $L = 10$, so we obtain 50 models from an optimizer. The computational budget for each optimizer (for one trajectory) is 300 epochs.

## OPTIMIZATION VERSUS GENERALIZATION

**Across Tasks** Our first analysis compares two sets of populations: SetA is the population of 50 lowest-loss models; for each trajectory, the 10 lowest-loss models are selected. In SetB (to which we refer as the oracle set earlier), from each trajectory, the 10 highest-accuracy (or highest macro-F1) models are selected. These two sets are compared in terms of accuracy (or macro-F1) via the two-sided t-test or the Mann-Whitney U test (we utilize `scipy.stats.ttest_ind` and `scipy.stats.mannwhitneyu`). Both test the null hypothesis that the distribution underlying SetA is the same as the distribution underlying SetB. While the t-test is parametric, the Mann-Whitney U test is nonparametric and so is a good choice when the data is not normally distributed and there are no ties (as opposed to the Wilcoxon test which makes both assumptions). In the interest of space, we only report (in Table 2) the Mann-Whitney U test for the hyperparameter-tuned algorithms here, relating the rest of the tests (and on the base algorithms) in the Supplementary Material. Table 2 shows that with few exceptions (NiM-SGD and NiG-BH on GoEmotions), the null hypothesis cannot be rejected. That is, one cannot reject that the distribution underlying SetA (models selected by loss) is the same as the distribution underlying SetB (models selected by test set accuracy/macro-F1). The results in the Supplementary Material support this finding. We repeat this analysis on the ImageNet task with ResNet50, a more computationally-expensive task. However, unlike other work, we do not start with a pre-trained model but with initial weights sampled uniformly at random so that we can truly evaluate the performance of optimizers over a population of models, limiting each trajectory to 50 epochs. On three representative optimizers, SGD, SAM, and NiM-BH the Mann-Whitney U test yields corresponding p-values of 0.11543, 0.0865, and 0.2481, all above the threshold needed to reject the null hypothesis, in agreement with our findings on other tasks.

**Across Model Architectures** In the Supplementary Material we expand the comparison over model architectures (ResNet18, ResNet32, ResNet100, Wide-ResNet (40×10), and PyramidNet for computer vision tasks and DistillBERT and RoBERTa for NLP tasks). Hypothesis testing shows that the null hypothesis cannot be rejected, and so our findings are not impacted by model architectures.

| Algorithm | CIFAR10 ResNet50 | CIFAR100 ResNet50 | GoEmotions BERT | TweetEval BERT |
|---|---|---|---|---|
| SGD | 0.0821 | 0.1941 | 0.4192 | 0.1359 |
| NiG-SGD | 0.4231 | 0.2519 | 0.3618 | 0.4532 |
| NiM-SGD | 0.17432 | 0.34121 | **0.03489** | 0.1837 |
| SAM | 0.0915 | 0.051783 | 0.2638 | 0.1834 |
| NiG-BH | 0.07532 | 0.6739 | **0.04868** | 0.4839 |
| NiM-BH | 0.18346 | 0.29734 | 0.18942 | 0.3574 |
| NiG-MpBH | 0.3164 | 0.09473 | 0.16389 | 0.3184 |
| NiM-MpBH | 0.08633 | 0.4532 | 0.37647 | 0.07465 |

Table 2: Mann-Whitney U test comparing SetA to SetB for each optimizer over each real-world task. P-values $< 0.05$ are highlighted in bold font.

In Table 3 we relate the average and standard deviation of the test accuracy or macro-F1 for SetA versus SetB for each optimizer. We focus on the hyperparameter-optimized optimizers. Box plots are related in the Supplementary Material. Comparison across optimizers over SetA and SetB reveals comparable accuracies and standard deviations. Interesting observations emerge. Focusing on SetA (low-loss models), we observe that on the accuracy-evaluated tasks, CIFAR10 (ResNet50) and CIFAR100 (ResNet 50), the top three optimizers (with highest three accuracies) are SGD (twice), NimSGD (once), SAM (twice), and NiM-MpBH (once). On the macro-F1-evaluated tasks, GoEmotions (BERT) and TweetEval (BERT), the top three optimizers (with highest three macro-F1s) are SGD (once), NiM-SGD (once), NiM-BH (twice), and NiG-MpBH (twice). The BH-based optimizers have a slight advantage over SAM on the macro-F1 tasks.

| Algorithm | CIFAR10 ResNet50 | CIFAR100 ResNet50 | GoEmotions BERT | TweetEval BERT |
|---|---|---|---|---|
| SGD | (**0.934**,0.004) (0.929,0.002) | (**0.776**, 0.021) (0.785, 0.021) | (0.493, 0.032) (0.501, 0.029) | (**0.599**, 0.025) (0.609, 0.019) |
| NiG-SGD | (0.915, 0.004) (0.918, 0.004) | (0.759, 0.029) (0.763, 0.018) | (0.485, 0.051) (0.482, 0.049) | (0.572 , 0.037) (0.579 , 0.032) |
| NiM-SGD | (0.917, 0.005) (0.925, 0.004) | (**0.779**, 0.027) (0.786, 0.018) | (**0.501**, 0.044) (0.509, 0.039) | (0.594, 0.029) (0.596, 0.028) |
| SAM | (**0.924**, 0.017) (0.941, 0.007) | (**0.779**, 0.037) (0.793, 0.015) | (0.459, 0.041) (0.482, 0.023) | (0.589, 0.037) (0.595, 0.017) |
| NiG-BH | (0.908, 0.005) (0.912, 0.003) | (0.743, 0.019) (0.753, 0.015) | (0.486, 0.042) (0.495, 0.036) | (0.579, 0.031) ( 0.581, 0.298) |
| NiM-BH | (0.896, 0.019) (0.903, 0.009) | (0.749, 0.024) (0.759, 0.022) | (**0.503**, 0.053) (0.506, 0.038) | (**0.602**, 0.035) (0.607, 0.032) |
| NiG-MpBH | (0.904, 0.014) (0.906, 0.003) | (0.759, 0.027) (0.769, 0.022) | (**0.494**, 0.033) (0.502, 0.016) | (**0.613**, 0.027) (0.619, 0.025) |
| NiM-MpBH | (**0.919**, 0.004) (0.926, 0.003) | (0.751, 0.019) (0.764, 0.016) | (0.488, 0.039) (0.498, 0.031) | (0.579, 0.038) (0.587, 0.029) |

Table 3: For each optimizer, we relate the average accuracy and standard deviation over SetA (top row) and SetB (bottom row) for each optimizer. '(, )' relates '(average, standard deviation)' over models in a set. On the NLP tasks, summary statistics are for macro-F1. In bold font we highlight the performance (average accuracy or macro-F1 over SetA) by the three top optimizers for a particular task.

Figure 2 shows the distributions of test accuracy obtained by SGD, SAM, and NiM-BH for the ImageNet task (ResNet50). We observe that the medians are all close to one another. The medians of SetA and SetB are closer to each-other for SAM and NiM-BH.

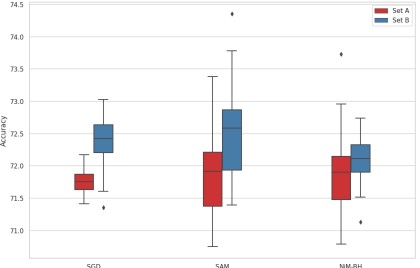

Figure 2: The distribution of test accuracy of SetA for SGD, SAM, and NiM-BH over ImageNet is in red. The distribution of SetB is in blue.

POPULATION-BASED COMPARISON OF OPTIMIZERS

We now compare pairs of optimizers. Instead of picking one model, we compare SetA of an optimizer to SetA of another optimizer. Recall that SetA is the population of low-loss models sampled by an optimizer over the loss landscape of a real-world task. These populations are compared on their test accuracy or macro-F1.

To test for differences between the resulting distributions, we utilize the Mann-Whitney U test with the null hypothesis that generalization (accuracy or macro-F1 depending on the task) in one group (Set A, optimizer X) is the same as the values in the other group (SetA, optimizer Y). Table 4 reports the p-values for SGD vs. SAM, SGD vs. NiM-BH, and SAM vs. NiM-BH. With two exceptions, the p-values are higher than $0.05$; the null hypothesis cannot be rejected. This suggests that when expanding

| Task | SGD vs. SAM | SGD vs. NiM-BH | SAM vs. NiM-BH |
|------|-------------|----------------|----------------|
| CIFAR10 | 0.3246 | 0.6542 | 0.0574 |
| CIFAR100 | 0.4745 | 0.1247 | **0.0458** |
| GoEmotions | **0.0355** | 0.1985 | 0.1749 |
| TweetEval | 0.2315 | 0.3254 | 0.2158 |
| ImageNet | 0.10227 | 0.1589 | 0.2857 |

Table 4: P-values are reported for the Mann Whitney U test of the null hypothesis that two distributions (test accuracies or macro-F1 over SetA) are the same. P-values $< 0.05$ are highlighted in bold font.

our view to a population of low-loss models obtained over several trajectories, we cannot distinguish in performance between SGD, SAM, and noise-enabled BH variants.

**Learning Curves** Figure 3 shows a learning curve (representative trajectory) for SGD, SAM, and NiM-BH. It is evident that SAM spends twice the number of gradient evaluations. If restricted to the same number of gradient evaluations as SGD and NiM-BH, SAM cannot reach low-loss regions.

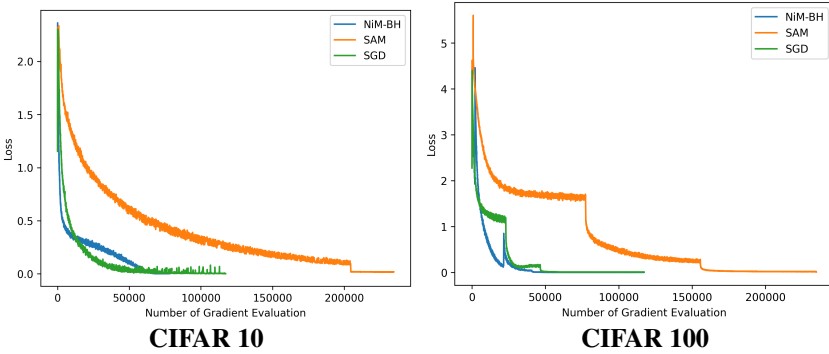

**CIFAR 10**    **CIFAR 100**

Figure 3: The learning curve for SGD, SAM, and NiM-BH. The y axis shows the smoothed loss (moving average), and the x-axis shows the gradient evaluations.

# 7 LIMITATIONS AND FUTURE WORK

While some attempt was made to optimize hyperparameters for each algorithm on a real-world task, this greatly increases the computational cost of benchmarking. In future work we plan to profile an increasing number of optimizers on more synthetic functions and real-world tasks, as well as study the impact of noise (magnitude of $\rho$ noise vector) in noise-enabled optimizers and its relationship with other hyperparameters for possibly combined effects on optimizer performance. In addition, noise-enabled optimizers and BH-based algorithms may provide interesting mechanisms to control for low loss, flatness, and other desired characteristics via which researchers can better understand and control for the relationship between better optimization and higher generalization capability.

# 8 CONCLUSION

In this paper we account for the inherent stochastic nature of SGD and noise-enabled variants. We introduce several optimizers under the BH framework. We propose a population-based approach to better characterize optimizers and improve our understanding of the relationship between optimization and generalization. The central insight we leverage is that during training an optimization trajectory grows in a nonconvex loss landscape, and so to characterize for the behavior of an optimizer one needs a nonlocal view that extends over several trajectories and goes beyond the "converged"/lowest-loss model. Our paper reveals several findings on the relationship between training loss and hold-out accuracy and the comparable performance of noise-enabled variants; indeed, these algorithms match the performance of flat-minima optimizers such as SAM with half the gradient evaluations. We hope this work will support further research in deep learning optimization relying not on single models but instead accounting for the stochasticity of optimizers.

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

PSEUDO-CODES OF NOISE IN GRADIENT AND NOISE IN MODEL METROPOLIS ALGORITHMS

Algorithm 10 shows the pseudocode for the NiG-MpBH algorithm, and Algorithm 10 shows the pseudocode for the NiM-MpBH algorithm. The main difference in these algorithms over the Monotonic BH versions shown in the main paper is that the monotonicity requirement is replaced with a probabilistic one, the Metropolis criterion. An additional user parameter, $\alpha$ is utilized for these algorithms. This parameter essentially determines how high of an increase in the loss function is allowed with some probability, as shown in lines 12 and 10, respectively. Decreases in loss are always accepted, but the Metropolis criterion allows the algorithm to allow temporary increases in loss to enhance its exploration probability and so increase the likelihood that better minima will be found further in the loss landscape. Please note that lines 16-18 and 14-16, respectively, are only essential for uses of these algorithms when the lowest-loss model is desired to be attracted. These lines are not essential. One typically monitors the training loss trajectory. In our particular setup in the main paper, we sample a fixed number of lowest-loss models from an optimization trajectory.

---

**Algorithm 10: NiG-MpBH**

1: **Input:** $f(\mathbf{w}), T > 0, \epsilon \cong 0, \tau > 0, \eta, \rho, \alpha$
2: **Output:** $\mathbf{w}_{best}$
3: $(\mathbf{w}, \Delta t) \leftarrow LclSearch(f, \mathbf{w}, \tau, \eta, \epsilon)$
4: $t \leftarrow t + \Delta t$
5: $\mathbf{w}_{best} \leftarrow \mathbf{w}$
6: **while** $t \leq T$ **do**
7: $\quad \mathbf{g} \leftarrow \nabla f(\mathbf{w}_t)$
8: $\quad \mathbf{g} \leftarrow PerturbGradient(\mathbf{g}, \rho)$
9: $\quad \mathbf{w} \leftarrow \mathbf{w} - \eta \cdot \mathbf{g}$
10: $\quad (\mathbf{w}_c, \Delta t) \leftarrow LclSearch(f, \mathbf{w}, \tau, \eta, \epsilon)$
11: $\quad \delta_f \leftarrow f(\mathbf{w}_c) - f(\mathbf{w})$
12: $\quad$ **if** $\delta_f < 0$ OR $exp(-\delta_f/\alpha) > rand(0, 1)$ **then**
13: $\quad\quad \mathbf{w} \leftarrow \mathbf{w}_c$
14: $\quad\quad t \leftarrow t + \Delta t$
15: $\quad$ **end if**
16: $\quad$ **if** $f(\mathbf{w}) < f(\mathbf{w}_{best})$ **then**
17: $\quad\quad \mathbf{w}_{best} \leftarrow \mathbf{w}$
18: $\quad$ **end if**
19: **end while**

---

$$\boxed{\begin{array}{l}
\hline
\qquad\qquad\qquad\text{Algorithm 11: NiM-MpBH}\\
\hline
\text{1: } \textbf{Input: } f(\mathbf{w}), T > 0, \epsilon \cong 0, \tau > 0, \eta, \rho, \alpha\\
\text{2: } \textbf{Output: } \mathbf{w}_{best}\\
\text{3: } (\mathbf{w}, \Delta t) \leftarrow LclSearch(f, \mathbf{w}, \tau, \eta, \epsilon)\\
\text{4: } t \leftarrow t + \Delta t\\
\text{5: } \mathbf{w}_{best} \leftarrow \mathbf{w}\\
\text{6: } \textbf{while } t \leq T \textbf{ do}\\
\text{7: } \quad \mathbf{w} \leftarrow PerturbModel(\mathbf{w}, \rho)\\
\text{8: } \quad (\mathbf{w}_c, \Delta t) \leftarrow Lcl(f, \mathbf{w}, \tau, \eta, \epsilon)\\
\text{9: } \quad \delta_f \leftarrow f(\mathbf{w}_c) - f(\mathbf{w})\\
\text{10: } \quad \textbf{if } \delta_f < 0 \text{ OR } exp(-\delta_f/\alpha) > rand(0,1) \textbf{ then}\\
\text{11: } \quad\quad \mathbf{w} \leftarrow \mathbf{w}_c\\
\text{12: } \quad\quad t \leftarrow t + \Delta t\\
\text{13: } \quad \textbf{end if}\\
\text{14: } \quad \textbf{if } f(\mathbf{w}) < f(\mathbf{w}_{best}) \textbf{ then}\\
\text{15: } \quad\quad \mathbf{w}_{best} \leftarrow \mathbf{w}\\
\text{16: } \quad \textbf{end if}\\
\text{17: } \textbf{end while}\\
\hline
\end{array}}$$

## LOCATIONS OF GLOBAL AND LOCAL MINIMA OF SYNTHETIC LOSS FUNCTIONS

We show performance on three selected synthetic functions in the main paper, but our evaluation considers six functions: Himmelblau (shown in the main paper), Three-Hump Camel (shown in the main paper), Six-Hump Camel (shown in the main paper), Beale, Rastrigin, and Rosenbrock. Below we show the contour plots for each of these functions as well as list their global and local minima if present.

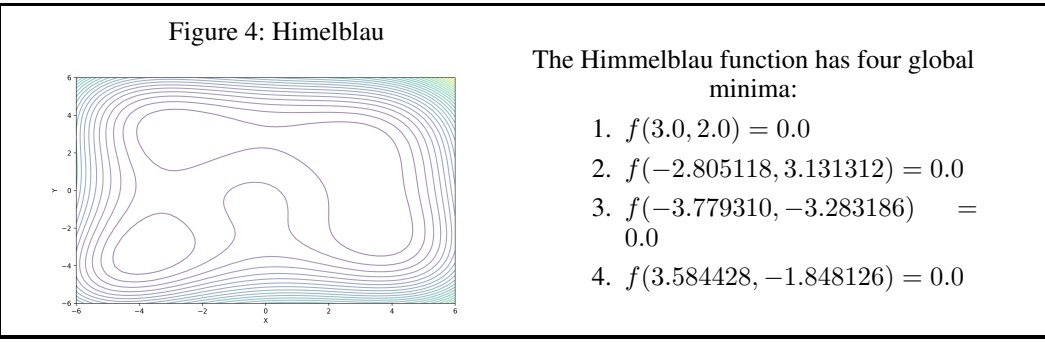

Figure 4: Himelblau

The Himmelblau function has four global minima:

1. $f(3.0, 2.0) = 0.0$
2. $f(-2.805118, 3.131312) = 0.0$
3. $f(-3.779310, -3.283186) = 0.0$
4. $f(3.584428, -1.848126) = 0.0$

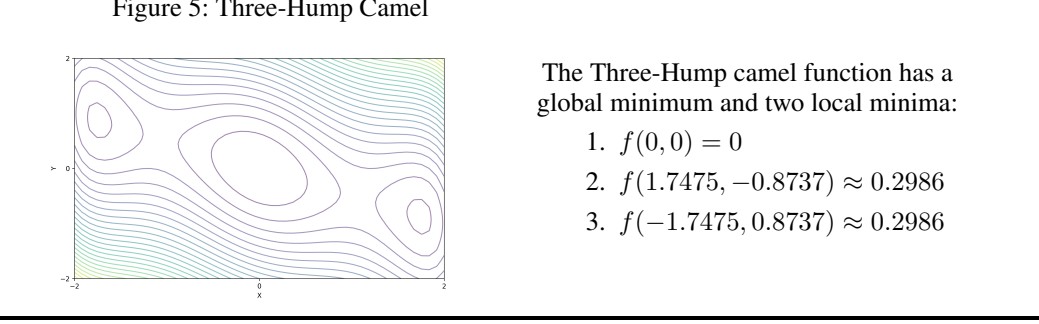

Figure 5: Three-Hump Camel

The Three-Hump camel function has a global minimum and two local minima:

1. $f(0, 0) = 0$
2. $f(1.7475, -0.8737) \approx 0.2986$
3. $f(-1.7475, 0.8737) \approx 0.2986$

| Figure 6: Six-Hump Camel | The Six-Hump Camel function has two global minima and four local minima: |
|---|---|
| 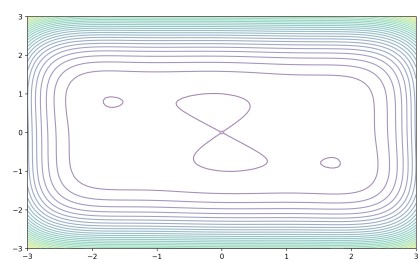 | 1. $f(-0.0898, 0.7126) = -1.0316$
2. $f(0.0898, -0.7126) = -1.0316$
3. $f(-2.8051, -0.0312) \approx 63.848$
4. $(0.9805, 1.8367) \approx -11.5$
5. $f(1.8839, -1.5252) \approx -3.14$
6. $f(-1.8658, 1.4900) \approx -2.64$ |

| Figure 7: Beale | |
|---|---|
| 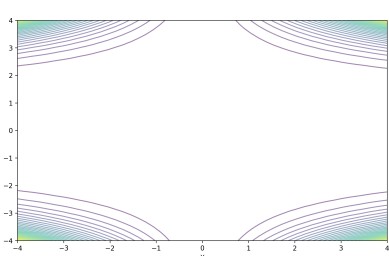 | The Beale function has one global minimum but a very broad plateau where optimization algorithms can get stuck (and many shallow local minima):

1. $f(3, 0.5) = 0$ |

| Figure 8: Rastrigin | The Rastrigin function has one global minimum and four local minima that are regularly distributed. |
|---|---|
| 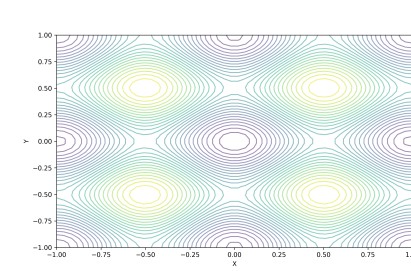 | 1. $f(-5.12, 5.12) = 529.537341$
2. $f(5.12, -5.12) = 529.537341$
3. $f(5.12, 5.12) = 529.537341$
4. $f(-5.12, -5.12) = 529.537341$
5. $f(0, 0) = 0$ |

| Figure 9: Rosenbrock | |
|---|---|
| 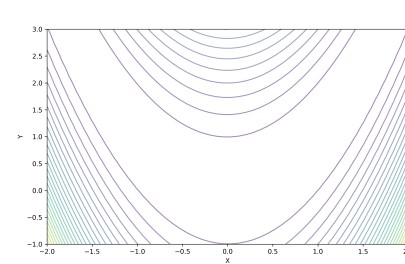 | The Rosenbrock function is a non-convex function. The global minimum is inside a long, narrow, parabolic-shaped flat valley. To find the valley is trivial, nut to converge to the global minimum, however, is difficult.

1. $f(1, 1) = 0$ |

We show here the stationary distribution for Beale, Rastrigin, and Rosenbrock functions (the main paper shows for the other three functions).

| Algorithms | Beale | | Rosenbrock | | Rastrigin | | | | | |
|---|---|---|---|---|---|---|---|---|---|---|
| | GM | Else | GM | Else | GM1 | LM1 | LM2 | LM3 | LM4 | Else |
| GD | 66 | 34 | 34 | 66 | 0 | 2.6 | 2.6 | 0 | 0 | 94.8 |
| NiG-GD | 68 | 32 | 30 | 70 | 0 | 5 | 6 | 0 | 0 | 89 |
| NiM-GD | 78 | 22 | **58** | 42 | 0 | 6.5 | 23.5 | 0 | 0 | 70 |
| SAM | 75 | 25 | **55** | 45 | 0 | 7 | 25 | 0 | 0 | **68** |
| NiG-BH | 77 | 23 | 42 | 58 | 0 | 7 | 24 | 0 | 0 | 69 |
| NiM-BH | 75 | 25 | **52** | 48 | 0 | 8.2 | 19.8 | 0 | 0 | 72 |
| NiG-MBH | **78** | 22 | 38 | 62 | 0 | 8 | 24 | 0 | 0 | **68** |
| NiM-MBH | 71 | 29 | 42 | 58 | 0 | 7.7 | 23.6 | 0 | 0 | 68.7 |
| NiG-MpBH | **79** | 58 | 42 | 58 | 0 | 6.3 | 24.0 | 0 | 0 | 69.7 |
| NiM-MpBH | **80** | 56 | 44 | 56 | 0 | 7.9 | 25.3 | 0 | 0 | **66.8** |

Table 5: The stationary distribution (reported in % for each entry) for the Beale, Rosenbrock, and Rastrigin function for each algorithm. The locations of the global minima (GM) and local minima (LM) for each function are listed above. The top three optimizers with the highest convergence to the global minimum on a given function are highlighted in bold font. For Rastrigin, where all optimizers have a very hard time converging to any minima, we highlight in bold font the top three optimizers that have the lowest percentage of end-points not converged to any of the minima (in the 'Else' category).

## VISUALIZATION OF STATIONARY DISTRIBUTIONS

Figure 10 shows 50 end-points (sampled from 500) of selected algorithms on three selected synthetic functions.

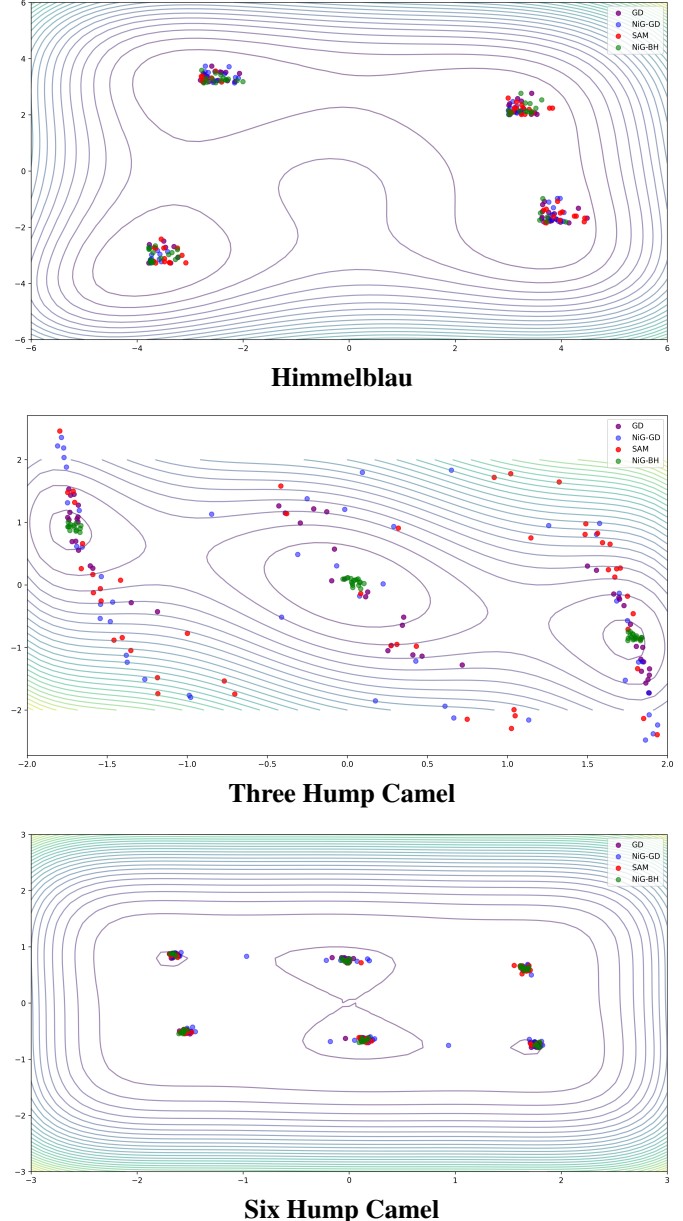

Figure 10: Stationary distribution of the optimization trajectory end-points by GD, NiG-GD (Jin et al., 2017), SAM (Foret et al., 2021), and NiG-BH. Distribution is shown for only 50 trajectories for each algorithm for a clear visual presentation.

We first compare SetA to SetB in terms of test set performance and then in terms of training loss.

## Comparing Test Set Performance

**Mann-Whitney U Test on Base Algorithms:** The Mann-Whitney-U test results on the hyperparameter-optimized algorithms are in the main paper. Table 6 reports the results on the base algorithms, with no hyperparameter optimization.

| Algorithm | CIFAR10 Resnet50 | CIFAR100 Resnet50 | GoEmotions | TweetEval |
|-----------|------------------|-------------------|------------|-----------|
| SGD | 0.2315 | 0.19332 | 0.2875 | 0.4621 |
| NiG-SGD | 0.2989 | 0.5429 | 0.4632 | 0.1654 |
| NiM-SGD | 0.6543 | 0.7563 | 0.6129 | 0.3219 |
| SAM | 0.0978 | **0.01073** | 0.1984 | 0.2861 |
| NiG-BH | 0.3569 | **0.0285** | 0.8328 | 0.3951 |
| NiM-BH | 0.6153 | **0.0472** | 0.6143 | 0.5178 |
| NiG-MpBH | 0.5421 | 0.1295 | 0.73256 | **0.01984** |
| NiM-MpBH | 0.6549 | 0.3219 | 0.2837 | 0.3542 |

Table 6: P-values are reported for the Mann-Whitney U test when comparing SetA to SetB for each algorithm over each of the real-world tasks. P-values less than $0.05$ are highlighted in bold font.

## T-Test on Hyperparameter-Optimized Algorithms

Table 7 reports results on t-tests on the hyperparameter-optimized algorithms. We test for the null hypothesis that two independent samples have identical average (expected) values. This test assumes that the populations have identical variances. With few exceptions, all p-values are under 0.05, so the null hypothesis cannot be rejected.

| Algorithm | CIFAR10 Resnet50 | CIFAR100 Resnet50 | GoEmotions | TweetEval |
|-----------|------------------|-------------------|------------|-----------|
| SGD | 0.2314 | 0.3156 | 0.7563 | 0.54623 |
| NiG-SGD | 0.4961 | 0.5753 | 0.72134 | 0.1823 |
| NiM-SGD | **0.0421** | 0.1291 | 0.2961 | 0.1962 |
| SAM | 0.7532 | 0.6982 | 0.6432 | 0.5391 |
| NiG-BH | 0.3612 | 0.18326 | **0.03135** | 0.1837 |
| NiM-BH | 0.6318 | 0.1938 | 0.7128 | 0.8723 |
| NiG-MpBH | 0.1834 | 0.7391 | 0.1935 | **0.02743** |
| NiM-MpBH | 0.13293 | 0.6254 | 0.5312 | 0.5193 |

Table 7: P-values are reported for the Mann-Whitney U test when comparing SetA to SetB for each algorithm over each of the real-world tasks. P-values less than $0.05$ are highlighted in bold font.

## T-test on Base Algorithms

Table 8 reports results on t-tests on the base versions of the algorithms (with no hyperparameter optimization). With few exceptions, all p-values are under 0.05, so the null hypothesis cannot be rejected.

## Comparing Training Loss

**Mann-Whitney U Test on Tuned Algorithms:** Table 9 reports this statistical test results on comparing the loss distributions corresponding to SetA and SetB for each of the (hyperparameter-optimized) algorithms/optimizers. With few exceptions, all p-values are under 0.05, so the null hypothesis cannot be rejected; that is, there are no statistically-significant differences between SetA and SetB in terms of loss, either.

**T-Test on Tuned Algorithms:** Table 10 reports this statistical test results on comparing the loss distributions corresponding to SetA and SetB for each of the (hyperparameter-optimized) algo-

| Algorithm | CIFAR10 Resnet50 | CIFAR100 Resnet50 | GoEmotions | TweetEval |
|---|---|---|---|---|
| SGD | 0.8764 | 0.14019 | 0.2345 | 0.5972 |
| NiG-SGD | 0.8195 | 0.8423 | 0.8744 | 0.4426 |
| NiM-SGD | 0.8345 | 0.7425 | 0.34556 | 0.4585 |
| SAM | 0.7213 | 0.76894 | 0.6754 | 0.6764 |
| NiG-BH | 0.5678 | **0.01245** | 0.45354 | 0.53254 |
| NiM-BH | 0.9134 | 0.8325 | 0.3958 | 0.6467 |
| NiG-MpBH | 0.6753 | 0.34869 | 0.6543 | 0.3958 |
| NiM-MpBH | 0.7423 | 0.38245 | 0.5649 | 0.2867 |

Table 8: P-values are reported for the Mann-Whitney U test when comparing SetA to SetB for each algorithm over each of the real-world tasks. P-values less than $0.05$ are highlighted in bold font.

| Algorithm | CIFAR10 Resnet50 | CIFAR100 Resnet50 | GoEmotions | TweetEval |
|---|---|---|---|---|
| SGD | 0.2164 | **0.0021** | 0.2123 | 0.2952 |
| NiG-SGD | 0.092 | 0.2385 | 0.0615 | 0.3152 |
| NiM-SGD | 0.1574 | 0.0612 | 0.1286 | 0.2032 |
| SAM | **0.0001** | **0.0048** | 0.3810 | 0.2357 |
| NiG-BH | **0.02858** | 0.1426 | **0.0318** | 0.5412 |
| NiM-BH | 0.3745 | 0.1854 | **0.0325** | 0.2548 |
| NiG-MpBH | **0.0345** | **0.0238** | 0.3740 | 0.2145 |
| NiM-MpBH | **0.00141** | **0.0217** | 0.1865 | 0.5402 |

Table 9: P-values are reported for the Mann-Whitney U test when comparing the loss distributions of SetA to SetB for each algorithm over each of the real-world tasks. P-values less than $0.05$ are highlighted in bold font.

rithms/optimizers. With few exceptions, all p-values are under 0.05, so the null hypothesis cannot be rejected.

| Algorithm | CIFAR10 Resnet50 | CIFAR100 Resnet50 | GoEmotions | TweetEval |
|---|---|---|---|---|
| SGD | 0.5631 | 0.7415 | 0.3534 | 0.1983 |
| NiG-SGD | 0.6512 | 0.1853 | **0.0916** | 0.4325 |
| NiM-SGD | 0.3259 | 0.7122 | 0.3214 | **0.0851** |
| SAM | **0.0015** | **0.0384** | 0.1120 | 0.2352 |
| NiG-BH | 0.1523 | 0.2854 | 0.3214 | **0.0254** |
| NiM-BH | 0.2145 | 0.3847 | **0.0978** | 0.3021 |
| NiG-MpBH | 0.1854 | **0.0631** | 0.2654 | 0.5546 |
| NiM-MpBH | 0.2541 | **0.0361** | 0.3845 | 0.4153 |

Table 10: P-values are reported for the T test when comparing the loss distributions of SetA to SetB for each algorithm over each real-world task. P-values less than $0.05$ are highlighted in bold font.

Figures 11-14 show the distribution of performance on the held-out test set of the 50 models selected by loss (to which we refer as SetA in the main manuscript) for each of the optimizers on each of the real-world tasks. On CIFAR10 (RestNet50) and CIFAR100 (ResNet50) the metric of performance is test set accuracy. On GoEmotions (BERT) and TweetEval (BERT) the metric of performance is macro-F1. Figures 15-18 do so for SetB.

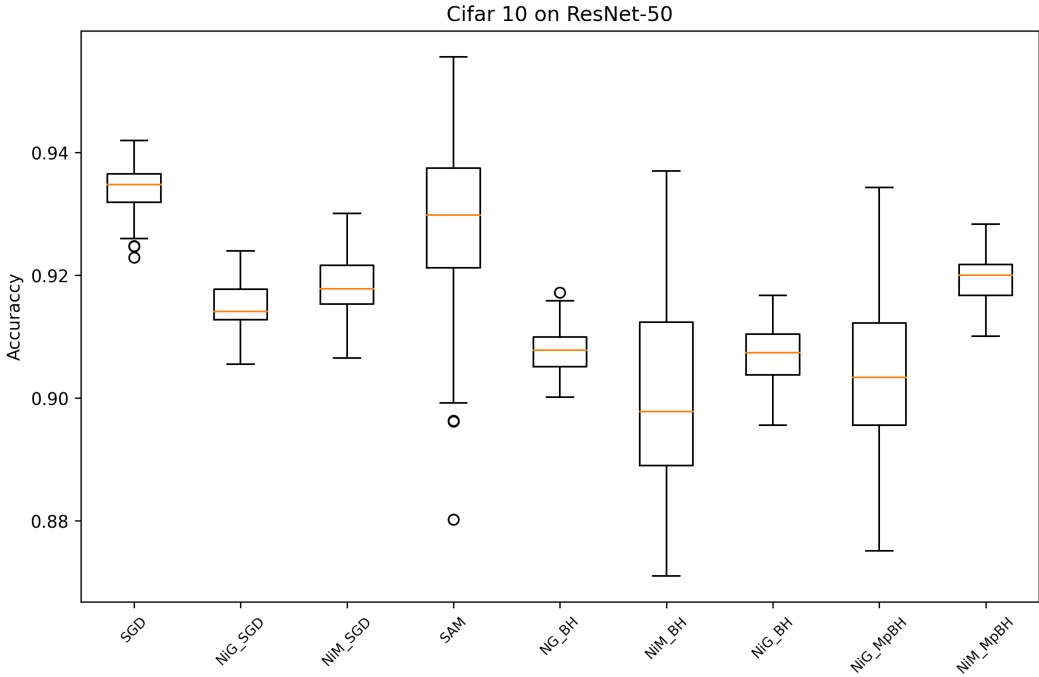

Figure 11: The distribution of test set accuracy of the 50 models extracted from each optimizer based on low-loss (to which we refer as SetA in the main paper) is shown here for each optimizer for the CIFAR10 (ResNet50) task.

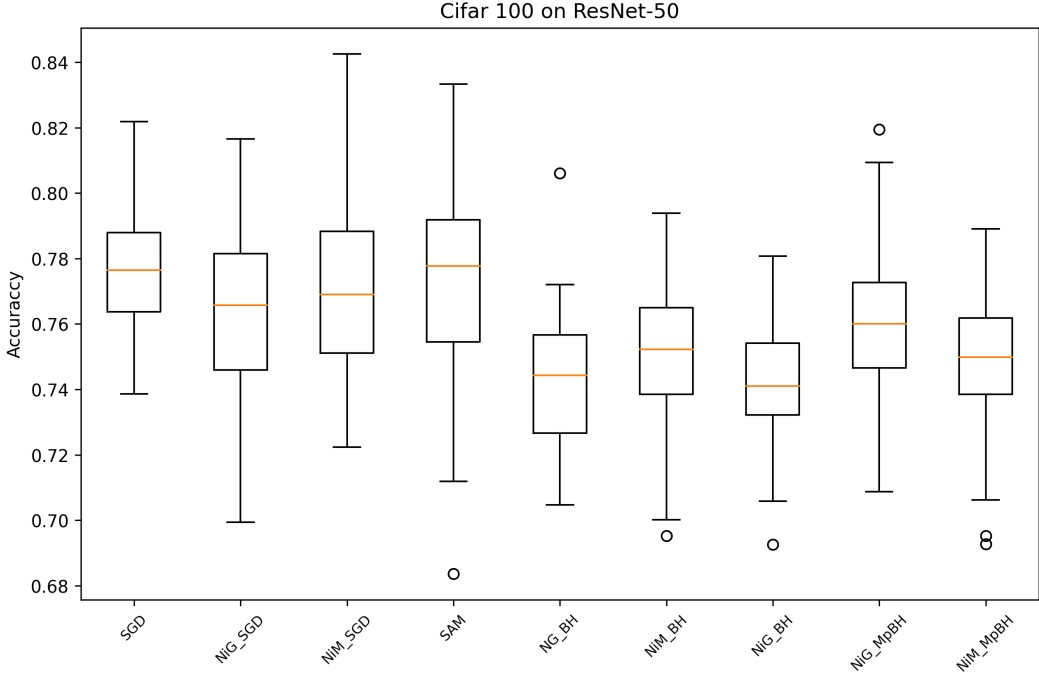

Figure 12: The distribution of test set accuracy of the 50 models extracted from each optimizer based on low-loss (to which we refer as SetA in the main paper) is shown here for each optimizer for the CIFAR100 (ResNet50) task.

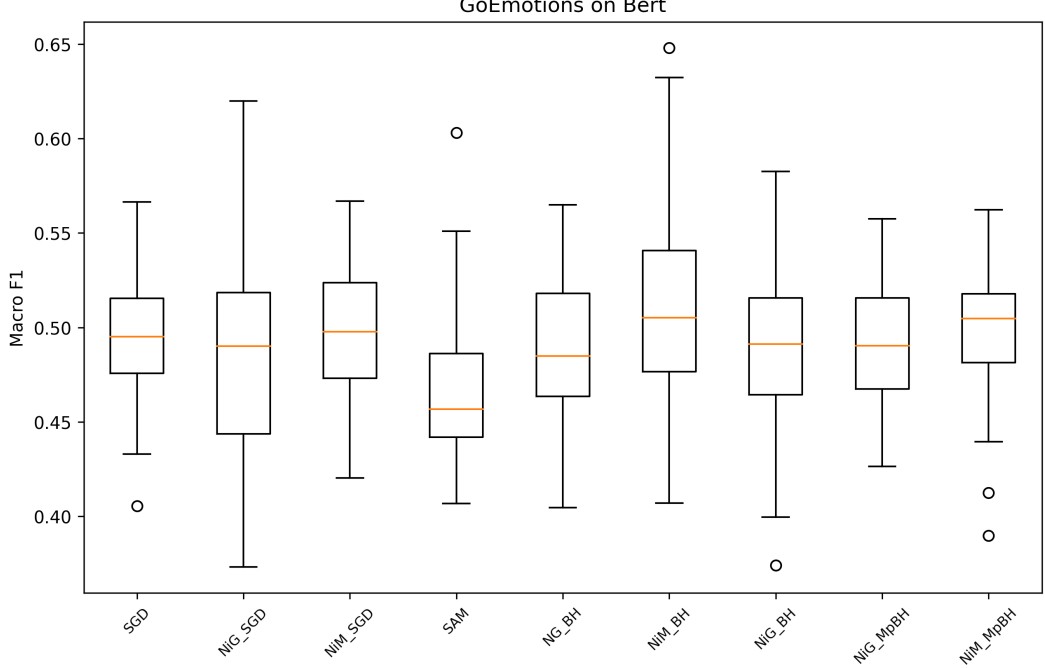

Figure 13: The distribution of macro-F1 score of the 50 models extracted from each optimizer based on low-loss (to which we refer as SetA in the main paper) is shown here for each optimizer for the GoEmotions (BERT) task.

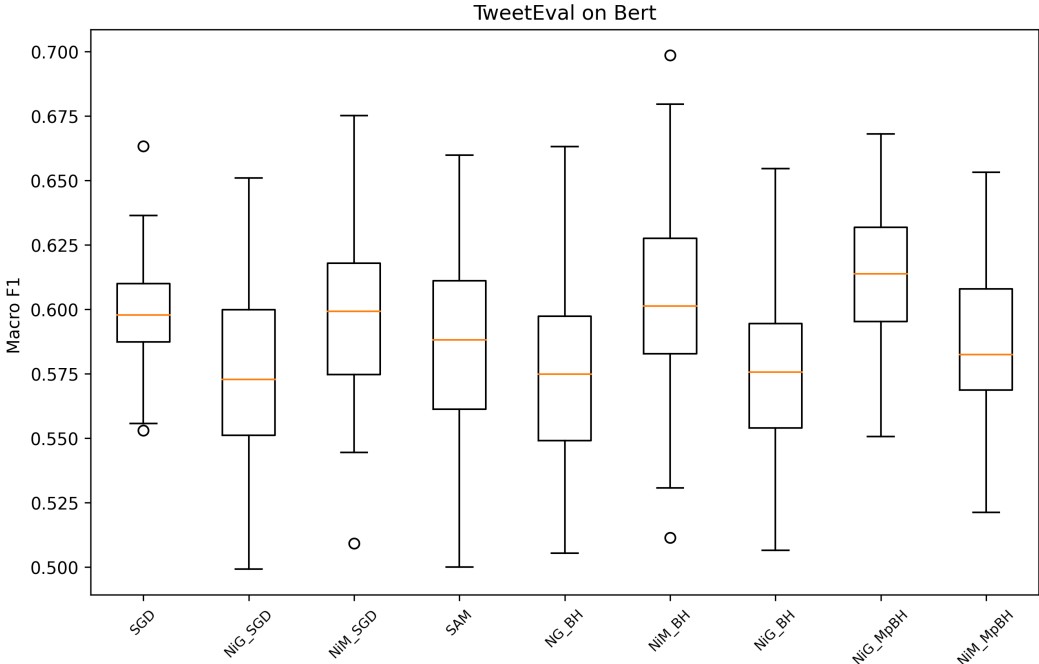

Figure 14: The distribution of macro-F1 score of the 50 models extracted from each optimizer based on low-loss (to which we refer as SetA in the main paper) is shown here for each optimizer for the TweetEval (BERT) task.

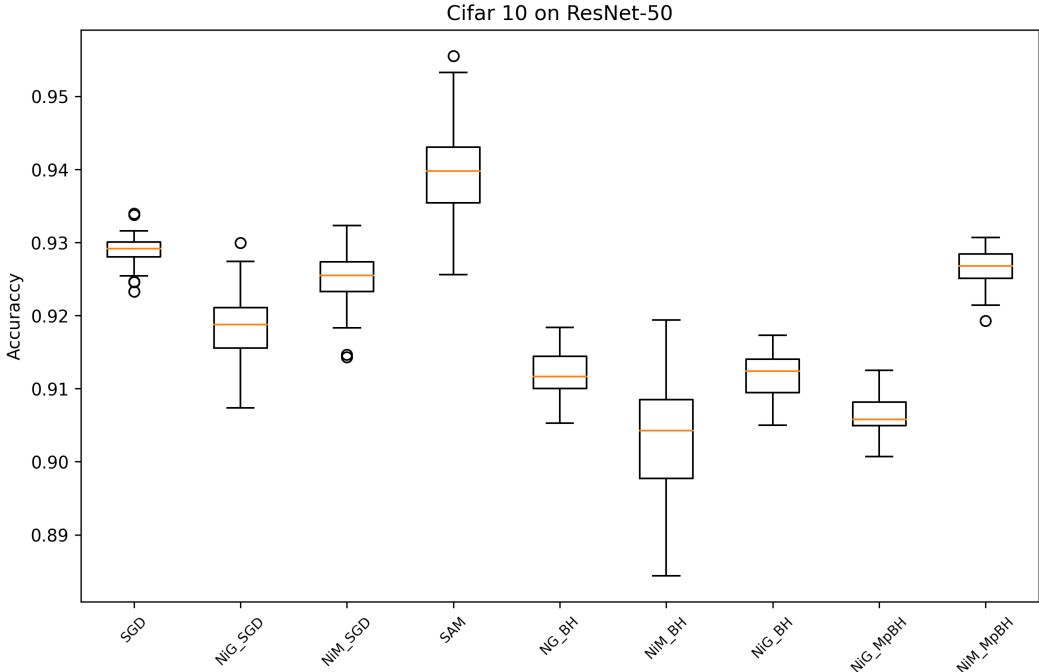

Figure 15: The distribution of test set accuracy of the 50 models extracted from each optimizer based on test set accuracy (to which we refer as SetB in the main paper) is shown here for each optimizer for the CIFAR10 (ResNet50) task.

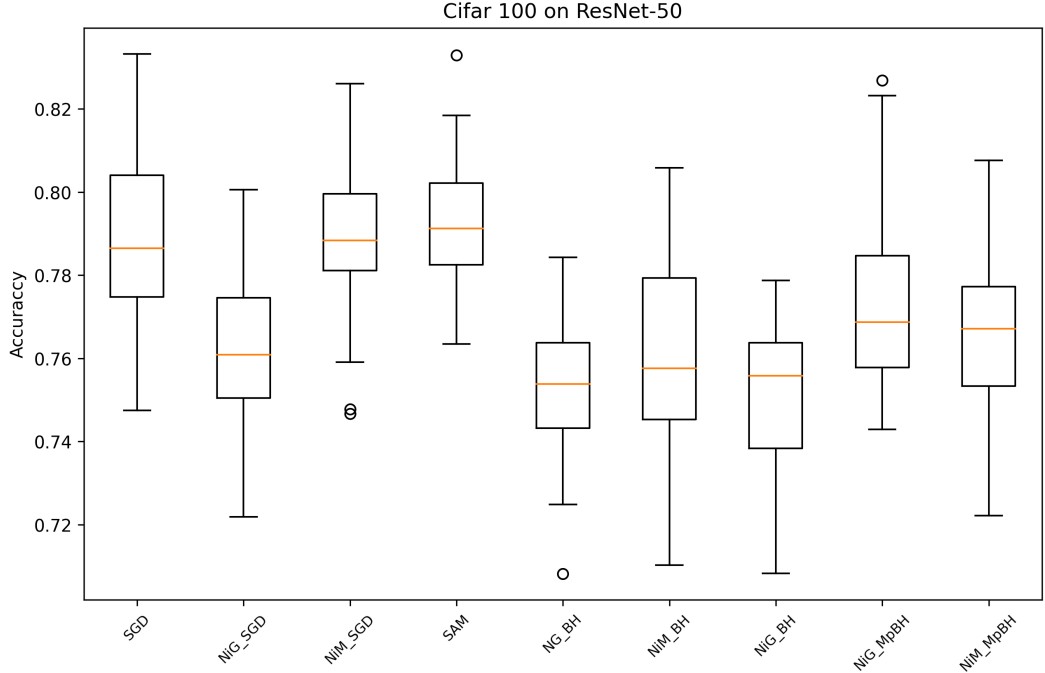

Figure 16: The distribution of test set accuracy of the 50 models extracted from each optimizer based on test set loss (to which we refer as SetB in the main paper) is shown here for each optimizer for the CIFAR100 (ResNet50) task.

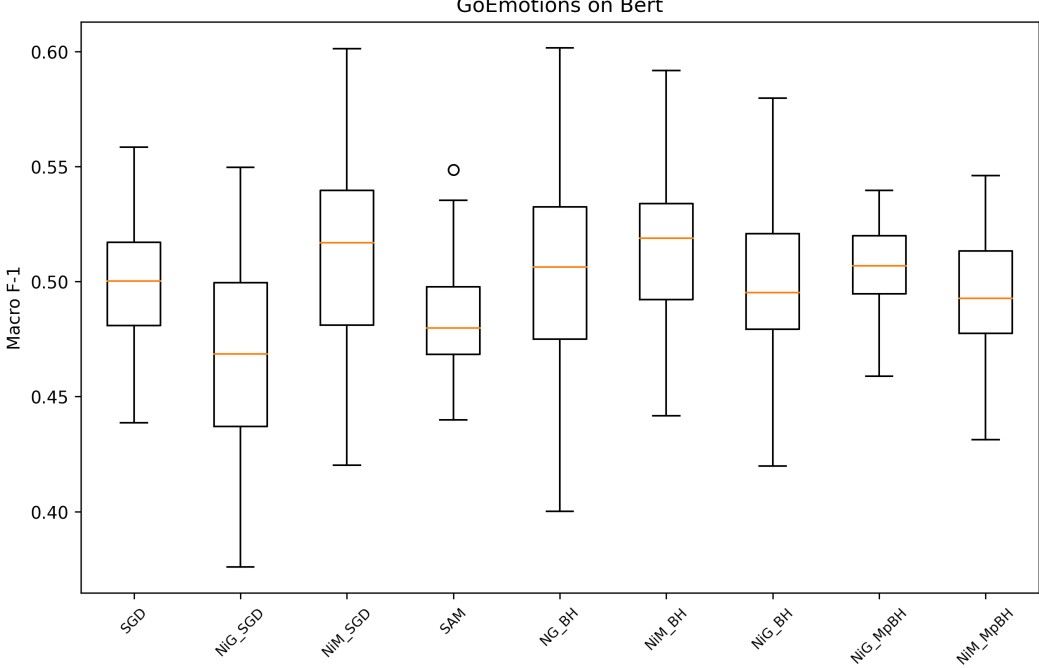

Figure 17: The distribution of test set macro-F1 of the 50 models extracted from each optimizer based on test set macro-F1 (to which we refer as SetB in the main paper) is shown here for each optimizer for the GoEmotions (BERT) task.

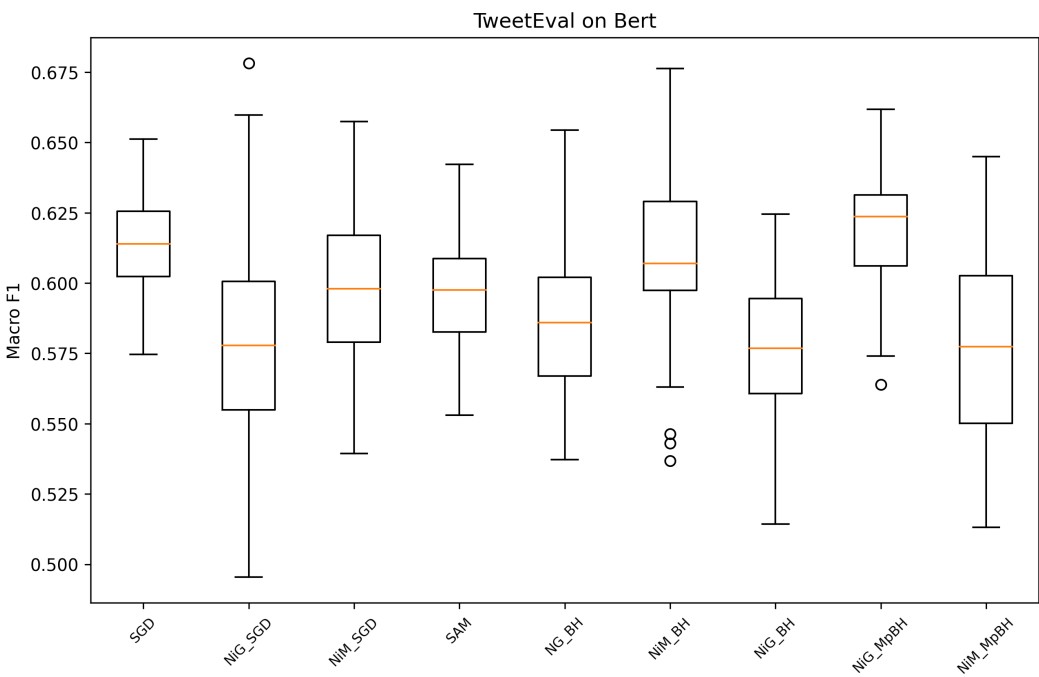

Figure 18: The distribution of test set macro-F1 of the 50 models extracted from each optimizer based on test set macro-F1 (to which we refer as SetB in the main paper) is shown here for each optimizer for the TweetEval (BERT) task.

Figures 19-22 show the distribution of performance on the training loss of the 50 models selected by loss (to which we refer as SetA in the main manuscript) for each of the optimizers on each of the real-world tasks. Figures 23-26 do so for SetB.

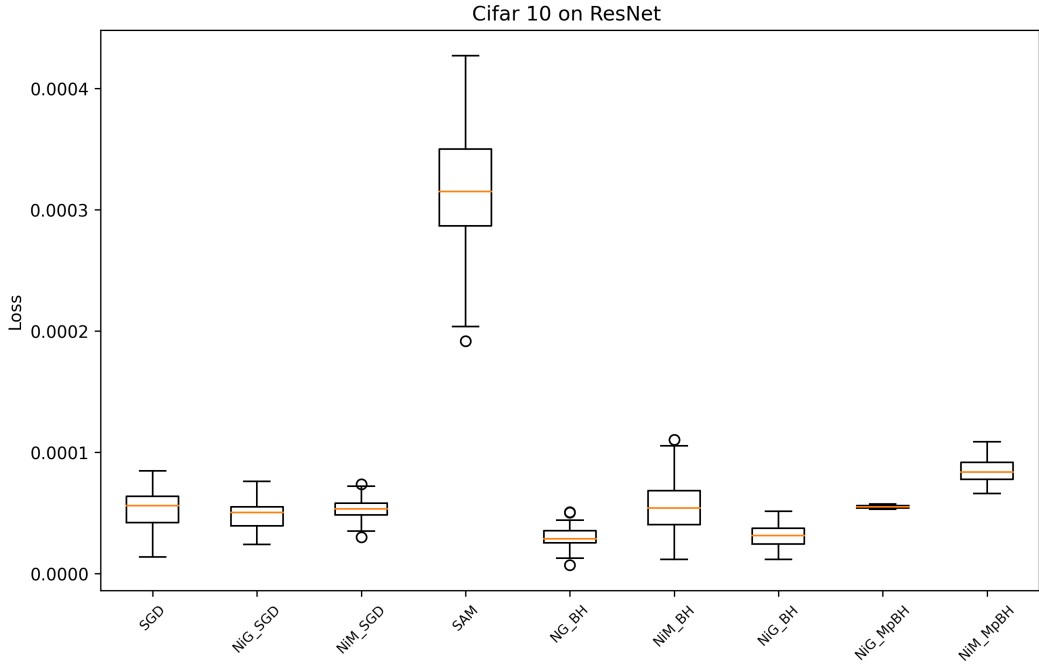

Figure 19: The distribution of training set loss of the 50 models extracted from each optimizer based on training set loss (to which we refer as SetA in the main paper) is shown here for each optimizer for the CIFAR 10 (ResNet50) task.

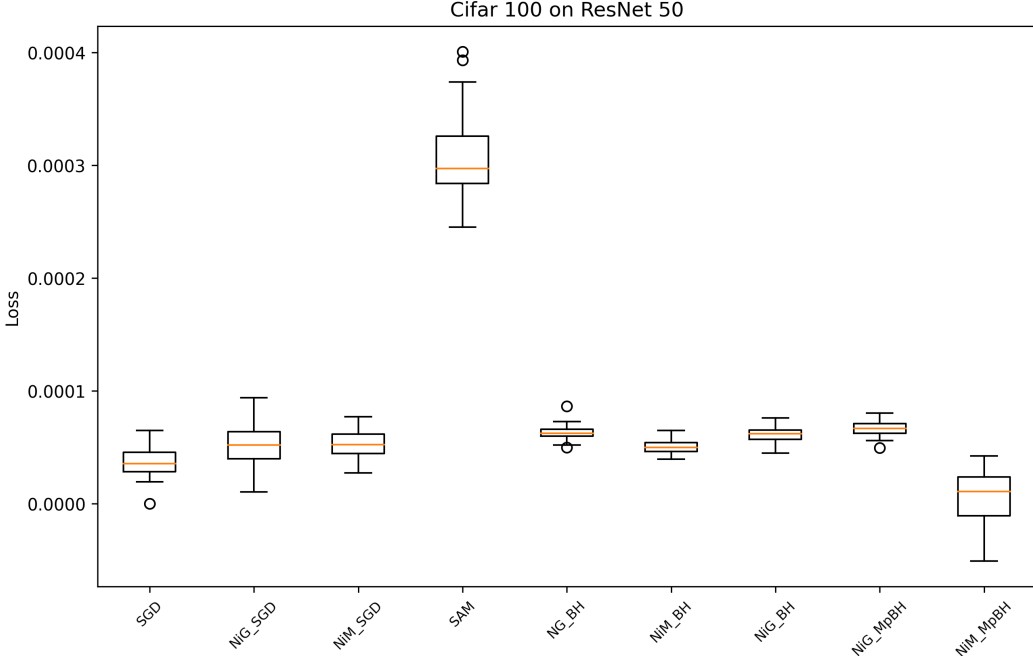

Figure 20: The distribution of training set loss of the 50 models extracted from each optimizer based on training set loss (to which we refer as SetA in the main paper) is shown here for each optimizer for the CIFAR 100 (ResNet50) task.

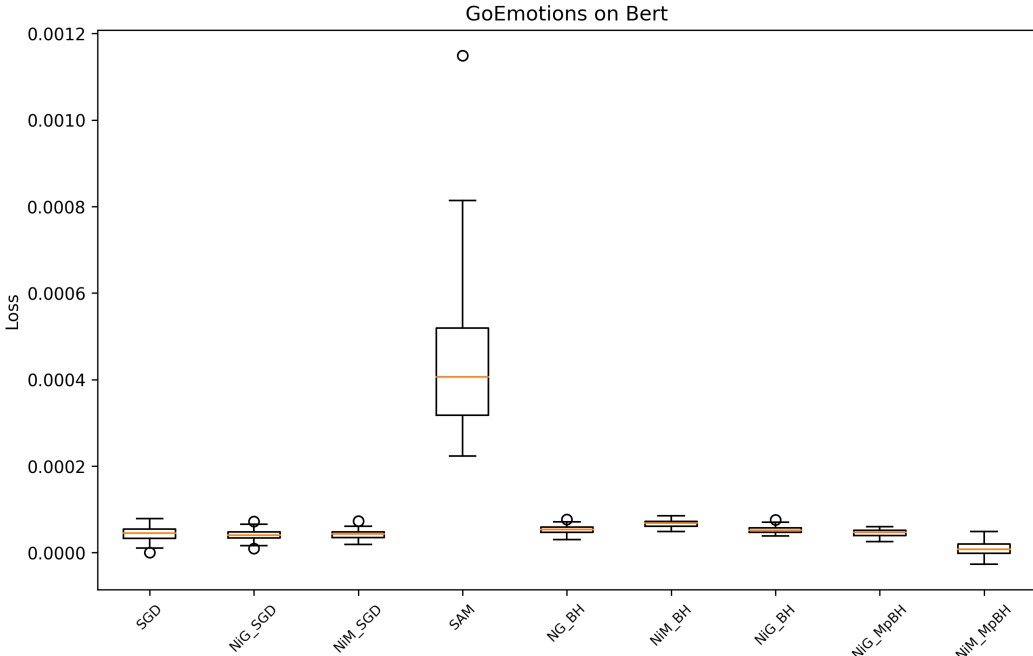

Figure 21: The distribution of training set loss of the 50 models extracted from each optimizer based on training set loss (to which we refer as SetA in the main paper) is shown here for each optimizer for the GoEmotions (BERT) task.

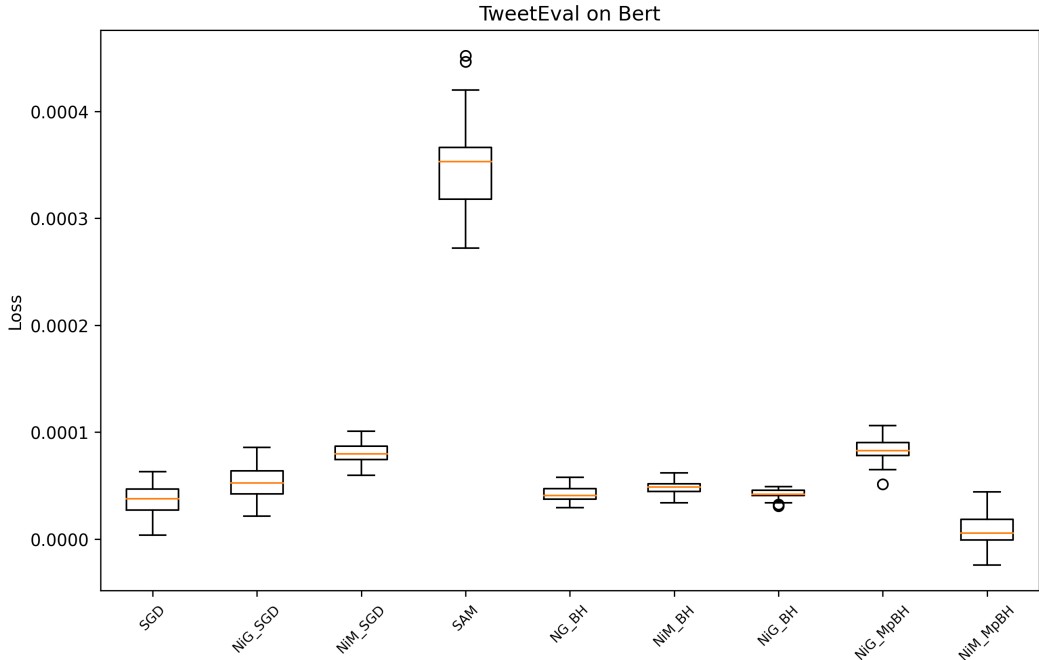

Figure 22: The distribution of training set loss of the 50 models extracted from each optimizer based on training set loss (to which we refer as SetA in the main paper) is shown here for each optimizer for the TweetEval (BERT) task.

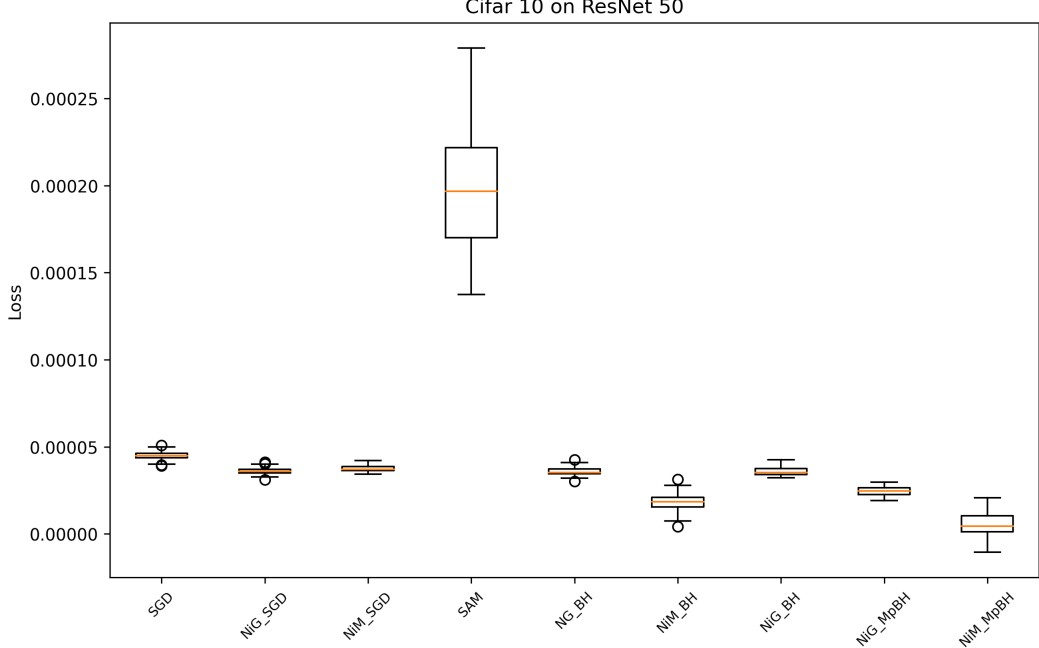

Figure 23: The distribution of training set loss of the 50 models extracted from each optimizer based on test set performance (to which we refer as SetB in the main paper) is shown here for each optimizer for the CIFAR 10 (ResNet50) task.

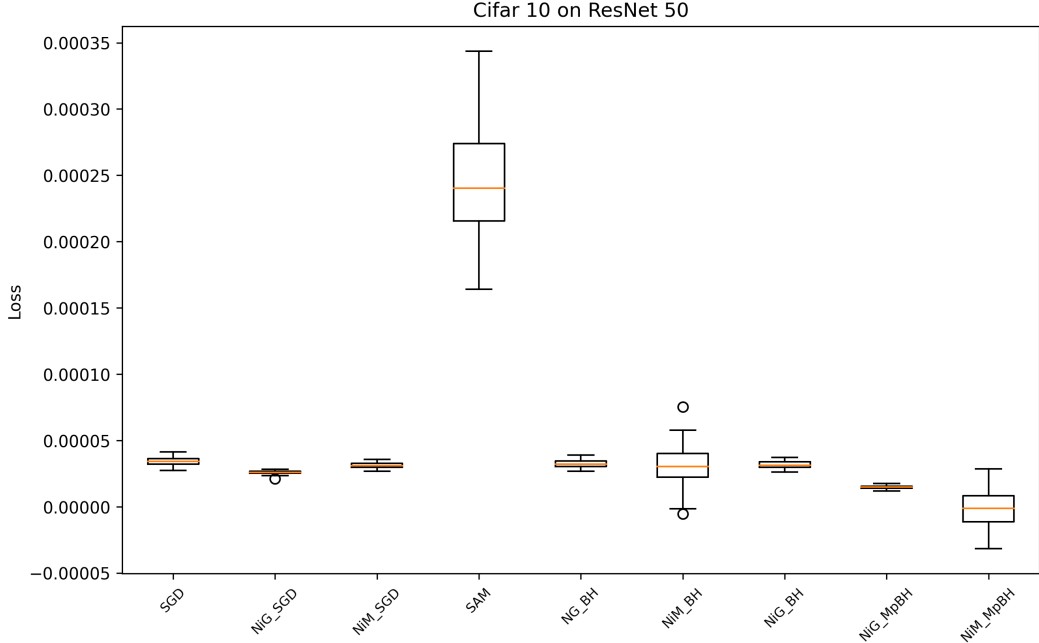

Figure 24: The distribution of training set loss of the 50 models extracted from each optimizer based on test set performance (to which we refer as SetB in the main paper) is shown here for each optimizer for the CIFAR 100 (ResNet50) task.

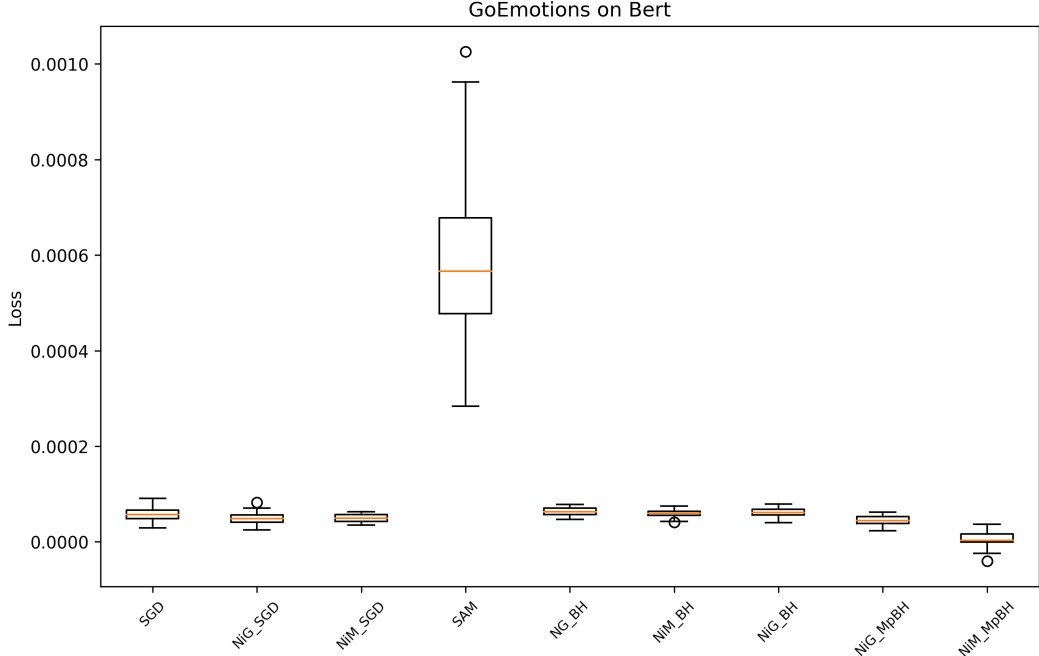

Figure 25: The distribution of training set loss of the 50 models extracted from each optimizer based on test set performance (to which we refer as SetB in the main paper) is shown here for each optimizer for the GoEmotions (Bert) task.

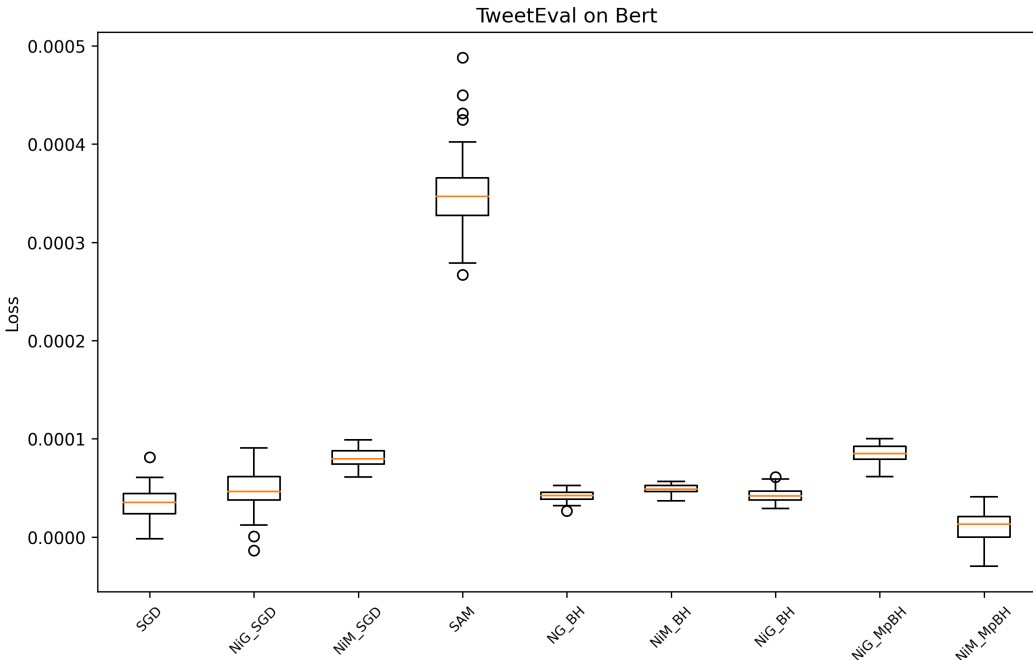

Figure 26: The distribution of training set loss of the 50 models extracted from each optimizer based on test set performance (to which we refer as SetB in the main paper) is shown here for each optimizer for the TwwetEval(Bert) task.

EXPANDED ANALYSIS ON VARIOUS ARCHITECTURES

We expand the analysis presented in the main paper to different architectures. For a given optimizer, for each world task and for a particular model architecture, we compare SetA to SetB to relate between optimization and generalization in the statistical sense. Tables below report p-values obtained with the Mann-Whitney U test. In the main paper we report analysis with ResNet50 for CIFAR10 and CIFAR100. Here we expand to consider ResNet18, ResNet32, ResNet100, Wide-ResNet (40 X 10), and PyramidNet. In the main paper we report analysis with BERT for the NLP tasks GoEmotions and TweetEval. Here we include DistillBERT and RoBERTa. Hypothesis testing shows that the null hypothesis cannot be rejected, and so our findings are not impacted by different model architectures.

We report one table per optimizer in the interest of clarity. With few exceptions, all p-values are under 0.05, so the null hypothesis cannot be rejected. That is, the results reported in the main paper extend over model architectures, as well.

| Architecture | Problems | | | |
|---|---|---|---|---|
| | CIFAR 10 | CIFAR 100 | GoEmotions | TweetEval |
| ResNet 18 | 0.2514 | 0.1635 | - | - |
| ResNet 32 | 0.1164 | 0.0954 | - | - |
| ResNet 100 | 0.0962 | 0.1535 | - | - |
| Wide-Resnet (40 X 10 ) | 0.1824 | 0.0759 | - | - |
| PyramidNet | 0.0658 | 0.2975 | - | - |
| DistilBERT | - | - | 0.07645 | **0.03241** |
| RoBERTa | - | - | **0.04211** | 0.07531 |

Table 11: Mann-Whitney U test comparing SetA to SetB for SGD over each real-world task over several model architectures. P-values < 0.05 are highlighted in bold font.

| Architecture | Problems | | | |
|---|---|---|---|---|
| | CIFAR 10 | CIFAR 100 | GoEmotions | TweetEval |
| ResNet 18 | 0.2154 | 0.1639 | - | - |
| ResNet 32 | 0.08751 | 0.06834 | - | - |
| ResNet 100 | 0.1134 | 0.0861 | - | - |
| Wide-Resnet (40 X 10 ) | 0.2159 | 0.0618 | - | - |
| PyramidNet | **0.01534** | 0.0615 | - | - |
| DistilBERT | - | - | 0.2317 | 0.1851 |
| RoBERTa | - | - | 0.0517 | **0.04531** |

Table 12: Mann-Whitney U test comparing SetA to SetB for SAM over each real-world task. P-values < 0.05 are highlighted in bold font.

| Architecture | Problems | | | |
|---|---|---|---|---|
| | CIFAR 10 | CIFAR 100 | GoEmotions | TweetEval |
| ResNet 18 | 0.1854 | 0.1125 | - | - |
| ResNet 32 | 0.09645 | 0.1531 | - | - |
| ResNet 100 | 0.2231 | 0.06543 | - | - |
| Wide-Resnet (40 X 10 ) | **0.0314** | 0.1741 | - | - |
| PyramidNet | 0.3129 | 0.05213 | - | - |
| DistilBERT | - | - | 0.1692 | **0.0414** |
| RoBERTa | - | - | **0.0134** | 0.0951 |

Table 13: Mann-Whitney U test comparing SetA to SetB for NiG-SGD over each real-world task. P-values < 0.05 are highlighted in bold font.

| Architecture | Problems | | | |
|---|---|---|---|---|
| | CIFAR 10 | CIFAR 100 | GoEmotions | TweetEval |
| ResNet 18 | 0.1642 | 0.0851 | - | - |
| ResNet 32 | 0.0613 | 0.3142 | - | - |
| ResNet 100 | 0.0832 | 0.1325 | - | - |
| Wide-Resnet (40 X 10 ) | 0.1751 | 0.2162 | - | - |
| PyramidNet | 0.0835 | 0.1923 | - | - |
| DistilBERT | - | - | 0.9761 | 0.0856 |
| RoBERTa | - | - | 0.0873 | 0.2143 |

Table 14: Mann-Whitney U test comparing SetA to SetB for NiM-SGD over each real-world task. P-values $< 0.05$ are highlighted in bold font.

| Architecture | Problems | | | |
|---|---|---|---|---|
| | CIFAR 10 | CIFAR 100 | GoEmotions | TweetEval |
| ResNet 18 | 0.1672 | **0.0332** | - | - |
| ResNet 32 | 0.3511 | 0.0867 | - | - |
| ResNet 100 | 0.1845 | 0.2071 | - | - |
| Wide-Resnet (40 X 10 ) | 0.1172 | **0.0313** | - | - |
| PyramidNet | 0.2512 | 0.1102 | - | - |
| DistilBERT | - | - | 0.0983 | **0.02143** |
| RoBERTa | - | - | 0.1524 | **0.04531** |

Table 15: Mann-Whitney U test comparing SetA to SetB for NiG-BH over each real-world task. P-values $< 0.05$ are highlighted in bold font.

| Architecture | Problems | | | |
|---|---|---|---|---|
| | CIFAR 10 | CIFAR 100 | GoEmotions | TweetEval |
| ResNet 18 | 0.0751 | 0.0855 | - | - |
| ResNet 32 | 0.2137 | 0.1980 | - | - |
| ResNet 100 | 0.0631 | 0.1124 | - | - |
| Wide-Resnet (40 X 10 ) | 0.0923 | 0.2513 | - | - |
| PyramidNet | 0.0985 | **0.0245** | - | - |
| DistilBERT | - | - | **0.0213** | 0.2261 |
| RoBERTa | - | - | **0.0198** | 0.0678 |

Table 16: Mann-Whitney U test comparing SetA to SetB for NiM-BH over each real-world task. P-values $< 0.05$ are highlighted in bold font.

| Architecture | Problems | | | |
|---|---|---|---|---|
| | CIFAR 10 | CIFAR 100 | GoEmotions | TweetEval |
| ResNet 18 | 0.0763 | 0.1712 | - | - |
| ResNet 32 | 0.1870 | 0.0764 | - | - |
| ResNet 100 | 0.3129 | **0.0413** | - | - |
| Wide-Resnet (40 X 10 ) | 0.1321 | 0.0571 | - | - |
| PyramidNet | 0.1439 | 0.0591 | - | - |
| DistilBERT | - | - | 0.2254 | 0.0848 |
| RoBERTa | - | - | 0.1427 | 0.2185 |

Table 17: Mann-Whitney U test comparing SetA to SetB for NiG-MpBH over each real-world task. P-values $< 0.05$ are highlighted in bold font.

| Architecture | Problems | | | |
|---|---|---|---|---|
| | CIFAR 10 | CIFAR 100 | GoEmotions | TweetEval |
| ResNet 18 | 0.1293 | 0.0878 | - | - |
| ResNet 32 | 0.0587 | 0.1859 | - | - |
| ResNet 100 | 0.2391 | 0.1187 | - | - |
| Wide-Resnet (40 X 10 ) | 0.1781 | 0.0763 | - | - |
| PyramidNet | 0.1534 | 0.2871 | - | - |
| DistilBERT | - | - | 0.2143 | 0.0885 |
| RoBERTa | - | - | 0.0912 | 0.1065 |

Table 18: Mann-Whitney U test comparing SetA to SetB for NiM-MpBH over each real-world task. P-values $< 0.05$ are highlighted in bold font.

We now relate the generalization performance of SetA for each optimizer on each task on each model architecture. As in the main paper, the average accuracy (or macro-F1 for NLP tasks) and standard deviation are reported for SetA in each setting. These summary statistics are juxtaposed to the summary statistics over SetB in each setting.

| Architecture | Problems | | | |
|---|---|---|---|---|
| | CIFAR 10 | CIFAR 100 | GoEmotions | TweetEval |
| ResNet 18 | (0.930,0.012) | (0.753, 0.032) | - | - |
| | (0.911,0.004) | (0.762, 0.011) | - | - |
| ResNet 32 | (0.921, 0.003) | (0.759, 0.023) | - | - |
| | (0.932, 0.002) | (0.761, 0.015) | - | - |
| ResNet 100 | (0.947, 0.005) | ( 0.787, 0.022) | - | - |
| | (0.949, 0.0012) | (0.786, 0.018) | - | - |
| Wide-Resnet (40 X 10 ) | ( 0.967, 0.021) | ( 0.813, 0.028) | - | - |
| | (0.971, 0.019) | (0.819, 0.017) | - | - |
| PyramidNet-110 | (0.961, 0.005) | (0.812, 0.015) | - | - |
| | (0.971, 0.003) | (0.817, 0.013) | - | - |
| DistilBERT | - | - | (0.503, 0.053) | ( 0.602, 0.035) |
| | - | - | (0.506, 0.038) | (0.607, 0.327) |
| RoBERTa | - | - | (0.494, 0.033) | (0.613, 0.027) |
| | - | - | (0.502, 0.016) | (0.619, 0.025) |

Table 19: For each architecture, we relate the average accuracy and standard deviation over SetA (top row) and SetB (bottom row) for SGD. '(, )' relates '(average, standard deviation)' over models in a set. On the NLP tasks, summary statistics are for macro-F1.

| Architecture | Problems | | | |
|---|---|---|---|---|
| | CIFAR 10 | CIFAR 100 | GoEmotions | TweetEval |
| ResNet 18 | (0.911,0.015) | (0.761, 0.025) | - | - |
| | (0.909,0.005) | (0.759, 0.018) | - | - |
| ResNet 32 | (0.921, 0.018) | (0.769, 0.022) | - | - |
| | (0.927, 0.014) | (0.751, 0.008) | - | - |
| ResNet 100 | (0.945, 0.005) | 0.788, 0.017) | - | - |
| | (0.955, 0.004) | (0.791, 0.028) | - | - |
| Wide-Resnet (40 X 10 ) | 0.961, 0.011) | 0.8123, 0.027) | - | - |
| | (0.966, 0.012) | (0.823, 0.021) | - | - |
| PyramidNet | (0.965, 0.005) | (0.821 0.015) | - | - |
| | (0.971, 0.013) | (0.823, 0.011) | - | - |
| DistilBERT | - | - | ( 0.515, 0.032) | (0.611, 0.041) |
| | - | - | (0.521, 0.011) | (0.621, 0.012) |
| RoBERTa | - | - | (0.512, 0.013) | (0.621, 0.019) |
| | - | - | (0.525, 0.026) | (0.626, 0.031) |

Table 20: For each architecture, we relate the average accuracy and standard deviation over SetA (top row) and SetB (bottom row) for SAM. '(, )' relates '(average, standard deviation)' over models in a set. On the NLP tasks, summary statistics are for macro-F1.

| Architecture | Problems | | | |
|---|---|---|---|---|
| | CIFAR 10 | CIFAR 100 | GoEmotions | TweetEval |
| ResNet 18 | (0.912,0.012) | (0.756, 0.018) | - | - |
| | (0.909,0.005) | (0.755, 0.011) | - | - |
| ResNet 32 | (0.921, 0.009) | (0.761, 0.018) | - | - |
| | (0.925, 0.004) | (0.763, 0.022) | - | - |
| ResNet 100 | (0.932, 0.015) | (0.781, 0.015) | - | - |
| | (0.945, 0.007) | (0.788, 0.021) | - | - |
| Wide-Resnet (40 X 10 ) | (0.955, 0.013) | (0.791, 0.021) | - | - |
| | (0.959, 0.004) | (0.795, 0.011) | - | - |
| PyramidNet | (0.961, 0.013) | (0.797, 0.011) | - | - |
| | (0.959, 0.009) | (0.787, 0.017) | - | - |
| DistilBERT | - | - | (0.511, 0.023) | (0.599, 0.021) |
| | - | - | (0.516, 0.026) | (0.601, 0.021) |
| RoBERTa | - | - | (0.512, 0.015) | (0.609, 0.005) |
| | - | - | (0.532, 0.006) | (0.611, 0.018) |

Table 21: For each architecture, we relate the average accuracy and standard deviation over SetA (top row) and SetB (bottom row) for NiG-SGD. '(, )' relates '(average, standard deviation)' over models in a set. On the NLP tasks, summary statistics are for macro-F1.

| Architecture | Problems | | | |
|---|---|---|---|---|
| | CIFAR 10 | CIFAR 100 | GoEmotions | TweetEval |
| ResNet 18 | (0.891,0.011) | 0.744, 0.025) | - | - |
| | (0.881,0.013) | (0.751, 0.012) | - | - |
| ResNet 32 | (0.901, 0.013) | (0.754, 0.022) | - | - |
| | (0.905, 0.002) | (0.758, 0.023) | - | - |
| ResNet 100 | (0.921, 0.015) | (0.768, 0.025) | - | - |
| | (0.918, 0.007) | (0.776, 0.015) | - | - |
| Wide-Resnet (40 X 10 ) | (0.925, 0.021) | (0.781, 0.029) | - | - |
| | (0.928, 0.011) | (0.788, 0.021) | - | - |
| PyramidNet | (0.944, 0.014) | (0.791, 0.019) | - | - |
| | (0.948, 0.011) | (0.798, 0.021) | - | ( - |
| DistilBERT | - | - | (0.511, 0.013) | (0.612,0.022) |
| | - | - | (0.521, 0.011) | (0.611, 0.029) |
| RoBERTa | - | - | (0.517, 0.009) | (0.609, 0.017) |
| | - | - | (0.511, 0.013) | (0.613, 0.023) |

Table 22: For each architecture, we relate the average accuracy and standard deviation over SetA (top row) and SetB (bottom row) for NiM-SGD. '(, )' relates '(average, standard deviation)' over models in a set. On the NLP tasks, summary statistics are for macro-F1.

| Architecture | Problems | | | |
|---|---|---|---|---|
| | CIFAR 10 | CIFAR 100 | GoEmotions | TweetEval |
| ResNet 18 | (0.921,0.007) | (0.761, 0.023) | - | - |
| | (0.924,0.006) | (0.768, 0.015) | - | - |
| ResNet 32 | (0.919, 0.011) | (0.773, 0.015) | - | - |
| | (0.922, 0.010) | (0.779, 0.012) | - | - |
| ResNet 100 | (0.941, 0.021) | 0.781, 0.019) | - | - |
| | (0.945, 0.011) | (0.787, 0.021) | - | - |
| Wide-Resnet (40 X 10 ) | (0.959, 0.014) | (0.801, 0.023) | - | - |
| | (0.964, 0.024) | (0.821, 0.021) | - | - |
| PyramidNet | (0.961, 0.027) | (0.833, 0.022) | - | - |
| | (0.962, 0.013) | (0.839, 0.028) | - | ( - |
| DistilBERT | - | - | (0.523, 0.023) | (0.622, 0.013) |
| | - | - | (0.541, 0.028) | (0.627, 0.013) |
| RoBERTa | - | - | (0.553, 0.026) | (0.63, 0.018) |
| | - | - | (0.552, 0.011) | (0.632, 0.010) |

Table 23: For each architecture, we relate the average accuracy and standard deviation over SetA (top row) and SetB (bottom row) for NiG-BH. '(, )' relates '(average, standard deviation)' over models in a set. On the NLP tasks, summary statistics are for macro-F1.

| Architecture | Problems | | | |
|---|---|---|---|---|
| | CIFAR 10 | CIFAR 100 | GoEmotions | TweetEval |
| ResNet 18 | ( 0.922,0.012) | (0.762,0.027) | - | - |
| | (0.929,0.012) | (0.771, 0.024) | - | - |
| ResNet 32 | (0.925, 0.012) | (0.769, 0.021) | - | - |
| | (0.928, 0.014) | (0.773, 0.021) | - | - |
| ResNet 100 | (0.947, 0.008) | (0.785, 0.016) | - | - |
| | (0.951, 0.022) | (0.786, 0.011) | - | - |
| Wide-Resnet (40 X 10 ) | ( 0.961, 0.007) | (0.811, 0.022) | - | - |
| | (0.969, 0.024) | (0.823, 0.005) | - | - |
| PyramidNet | (0.961, 0.028) | (0.831, 0.024) | - | - |
| | (0.965, 0.014) | (0.835, 0.029) | - | - |
| DistilBERT | - | - | (0.521, 0.022) | (0.632, 0.011) |
| | - | - | (0.533, 0.031) | (0.635 0.011) |
| RoBERTa | - | - | (0.544, 0.021) | (0.638,0.022) |
| | - | - | (0.552, 0.028) | (0.639, 0.021) |

Table 24: For each architecture, we relate the average accuracy and standard deviation over SetA (top row) and SetB (bottom row) for NiM-BH. '(, )' relates '(average, standard deviation)' over models in a set. On the NLP tasks, summary statistics are for macro-F1.

| Architecture | Problems | | | |
|---|---|---|---|---|
| | CIFAR 10 | CIFAR 100 | GoEmotions | TweetEval |
| ResNet 18 | (0.901,0.019) | (0.759,0.019) | - | - |
| | (0.911,0.017) | (0.764, 0.003) | - | - |
| ResNet 32 | (0.911, 0.010) | (0.771, 0.021) | - | - |
| | (0.915, 0.021) | (0.776, 0.014) | - | - |
| ResNet 100 | (0.934, 0.025) | (0.781, 0.021) | – | - |
| | (0.936, 0.011) | (0.787, 0.026) | - | - |
| Wide-Resnet (40 X 10 ) | (0.944, 0.011) | (0.80, 0.028) | - | - |
| | (0.949, 0.022) | (0.795 0.027) | - | - |
| PyramidNet | (0.954, 0.017) | (0.821, 0.009) | - | - |
| | (0.956, 0.017) | (0.811, 0.021) | - | - |
| DistilBERT | - | - | (0.521, 0.021) | (0.622,0.019) |
| | - | - | (0.527, 0.031) | (0.625, 0.027) |
| RoBERTa | - | - | (O.528, 0.014) | (0.625, 0.027) |
| | - | - | (0.531, 0.008) | (0.631, 0.032) |

Table 25: For each architecture, we relate the average accuracy and standard deviation over SetA (top row) and SetB (bottom row) for NiG-MpBH. '(, )' relates '(average, standard deviation)' over models in a set. On the NLP tasks, summary statistics are for macro-F1.

| Architecture | Problems | | | |
|---|---|---|---|---|
| | CIFAR 10 | CIFAR 100 | GoEmotions | TweetEval |
| ResNet 18 | ( 0.901,0.011) | (0.767,0.023) | - | - |
| | (0.905,0.010) | (0.765, 0.019) | - | - |
| ResNet 32 | (0.911, 0.021) | (0.767, 0.031) | - | - |
| | (0.921, 0.018) | (0.769, 0.020) | - | - |
| ResNet 100 | (0.925, 0.021) | (0.787, 0.031) | - | - |
| | (0.922, 0.008) | (0.786, 0.020) | - | - |
| Wide-Resnet (40 X 10 ) | (0.944, 0.028) | (0.809,0.022) | - | - |
| | (0.949, 0.011) | (0.809, 0.029) | - | - |
| PyramidNet | (0.957, 0.011) | (0.821, 0.027) | - | - |
| | (0.954, 0.026) | (0.833, 0.024) | - | - |
| DistilBERT | - | - | (0.521, 0.023) | (0.611,0.029) |
| | - | - | (0.533, 0.011) | (0.620, 0.021) |
| RoBERTa | - | - | (0.531, 0.026) | (0.622, 0.018) |
| | - | - | (0.534, 0.022) | (0.625, 0.026) |

Table 26: For each architecture, we relate the average accuracy and standard deviation over SetA (top row) and SetB (bottom row) for NiM-MpBH. '(, )' relates '(average, standard deviation)' over models in a set. On the NLP tasks, summary statistics are for macro-F1.

