# OpenReview forum: "Optimization and Generalizability: Fair Benchmarking for Stochastic Algorithms"
_ICLR.cc/2024/Conference — ICLR 2024 Conference Withdrawn Submission_

### Official Review · Reviewer_xXFv · 2023-10-29

**Soundness:** 4 excellent
**Presentation:** 3 good
**Contribution:** 3 good
**Rating:** 8
**Confidence:** 3

**Summary:**

This paper addresses the inherent stochastic nature of Stochastic Gradient Descent (SGD), its noise-enabled variants, and flat-minima optimizers. To understand these algorithms more deeply, there's a need to broaden the scope of noise-enabled SGD within the Basin Hopping framework. A central understanding is that during neural network training, the trajectory of optimization can be intricate, requiring a more comprehensive perspective than merely focusing on the converged or lowest-loss states.

The study introduces several novel algorithms, utilizing synthetic landscapes to rigorously evaluate the stationary distributions of various optimizers. Crucial findings reveal relationships between training loss, hold-out accuracy, and the performance of different optimizers, with some algorithms even matching the efficiency of flat-minima optimizers with half the gradient evaluations.

**Strengths:**

The paper is very well-written and easy to follow. This topic is of high interest to the community: given the ubiquity of SGD in daily machine learning practice, better understanding the intrinsic stochasticity of SGD in various loss landscapes is crucial.

The experiments carried out are both on synthetic data to gain intuition and on real-world data.

The context is correctly set and previous work duly cited.

**Weaknesses:**

On the content itself, I have nothing much to say. I think the research methodology is sound.

A minor details is the algorithm formatting in LaTex which could be enhanced. The instructions are colliding with the frame borders with almost no padding. This should be fixed for a nicer presentation.

**Questions:**

No question on my side.

---

> ### Author Response · Authors · 2023-11-22
>
> R: "The paper is very well-written and easy to follow. This topic is of high interest to the community: given the ubiquity of SGD in daily machine learning practice, better understanding the intrinsic stochasticity of SGD in various loss landscapes is crucial."
>
> A: We thank the reviewer for this perspective and summary of our work and for the kind words.
>
> R: "On the content itself, I have nothing much to say. I think the research methodology is sound."
>
> A:  We thank the reviewer for these encouraging words.
>
> R: "A minor details is the algorithm formatting in LaTex which could be enhanced. The instructions are colliding with the frame borders with almost no padding. This should be fixed for a nicer presentation."
>
> A: We thank the reviewer for prompting us to this issue. It was a side-effect of attempting to pack a lot of  content in the nine allowed pages. We have now increased the vertical and horizontal margins between the frame borders and the algorithms’ pseudocodes. We also tightened some of the variable names to allow for such margins, such as replacing “cand” with “c” in the superscript of “w” to indicate candidate weights, and replacing tls with “\Delta t” to indicate amount of time spent by the LocalSearch. Tightening up the language in several paragraphs also allowed us to better fit content in the allowed number of pages and accommodate larger font in the tables.

---

### Official Review · Reviewer_uLHr · 2023-11-01

**Soundness:** 2 fair
**Presentation:** 3 good
**Contribution:** 2 fair
**Rating:** 5
**Confidence:** 3

**Summary:**

This paper proposes a new method to benchmark the performance of an algorithm in terms of optimization and generalization. Instead of looking at the performance of a single trained model in the end, it proposes to form one group of models with lowest train loss and another group of models with best generalization performance by sampling from one or more trajectories. Then it can test whether the two groups come from the same population using various statistics such as t-test.  Besides this, it also comes up with a few new algorithms based on the Basin Hopping framework, and compare their performance with SGD, noise injected SGD (either through model or gradients), and sharpness-aware minimization (SAM) on synthetic and real-world datasets.

**Strengths:**

- Despite some typos, the paper can be easily understood. The related work section is very well-written, various algorithms and their connections are concisely summarized.

- The experiments on the synthetic dataset are interesting and reveal some useful information such as the distribution of SAM is skewed towards flatter minima.

**Weaknesses:**

- The proposed method may not be practical. First, if several models are sampled from one trajectory, they could be correlated, and the resulting statistical test may not be so useful. For example, with a bad initialization, the models sampled from the trajectory may not represent the population, which can lead to misinterprete the actual performance of the algorithm. Second, if multiple trajectories are required, this could increase the computation cost. Moreover, depending on the goal, one best model can be sufficient and a group of models is certainly not required.

- The experiments are not comprehensive. No large-scale datasets such as Imagenet are considered, and the neural network architecture is also limited.

**Questions:**

N/A

---

> ### Author Response · Authors · 2023-11-22
>
> R: "The experiments on the synthetic dataset are interesting and reveal some useful information such as the distribution of SAM is skewed towards flatter minima."
>
> A: We thank the reviewer for noting this.
>
> R: "The proposed method may not be practical. First, if several models are sampled from one trajectory, they could be correlated, and the resulting statistical test may not be so useful. For example, with a bad initialization, the models sampled from the trajectory may not represent the population, which can lead to misinterprete the actual performance of the algorithm. Second, if multiple trajectories are required, this could increase the computation cost. Moreover, depending on the goal, one best model can be sufficient and a group of models is certainly not required."
>
> A: The reviewer is correct that a trajectory gives you a biased view of the landscape. That is an important point we make in the manuscript. We argue for a population of models, hence the need to sample from multiple trajectories, which is what the paper does. The point on multiple trajectories coming at higher computational cost than one trajectory is correct, but we think it misses the point. We are not changing the behavior of any of the optimizers. We are not arguing for more expensive optimization algorithms. What we are trying to do in this paper is change the way we “do business.” The reviewer will note (and we make this point more formally in the paper) that the way much research that debuts a new optimization algorithms is the following: Here is our algorithm. Here is the converged model. Here is how the model performs on the held-out dataset on a variety of selected real-word problems. Some improvement in accuracy or macro-F1 is touted as the algorithm conferring better generalization. What we are arguing in this paper is that this won’t do in the presence of multi-dimensional nonconvex loss landscapes. Looking at one model ought to give us no confidence for why one would take this model and “deploy” it out there. Looking at one model also ought to give one no confidence for statements to the effect of “my algorithm is better than yours.” What is worse, many of these algorithms (including SGD) are stochastic, so again one cannot rely on one model. We argue that better clarity is provided when you switch the “benchmarking setup” from one model to a population of models. That is what we are attempting to do in this paper. The point on “depending on goal, one best model can be sufficient and a group of models is certainly not required” is somewhat chicken and egg. How would one know? Again, if you show that the population is homogeneous, that is great, but one would only know that if one introduces a nonlocal view in the first place, which is what we are arguing in this paper.
>
> R: "The experiments are not comprehensive. No large-scale datasets such as Imagenet are considered, and the neural network architecture is also limited."
>
> A: We had intended to add experiments on more real-world problems but ran out of time. We have now added ImageNet. Additionally, we had intended to add an important point on model architecture (e.g. ResNetX vs. ResNetY), which the reviewer astutely notes and which allows us to control in some manner for expected loss landscape complexity. We have added several model architectures both for the computer vision and the natural language processing tasks. Due to the limitation of space in the 9 pages in the main paper, we have added this analysis that additionally considers model architectures in the Supplementary Material, summarizing the main findings in the main paper.

---

### Official Review · Reviewer_eDnC · 2023-11-02

**Soundness:** 2 fair
**Presentation:** 1 poor
**Contribution:** 1 poor
**Rating:** 3
**Confidence:** 3

**Summary:**

The authors develop a collection of benchmarks, both with synthetic and real problems, and use them to compare SGD, SAM, and other noisy variants thereof, including:

1. SGD
2. Noise-in-gradient SGD
3. Noise-in-model SGD
4. SAM
5. Noise-in-gradient Basin Hopping
6. Noise-in-model Basin Hopping
7. Noise-in-gradient Metropolis Basin Hopping
8. Noise-in-model Metropolis Basin Hopping

These methods are each summarised before (and after) the benchmarks are described. Three synthetic objectives are considered: (I) Himmelblau, (II) Three Hump Camel, and (III) Six Hump Camel. Performance on these objectives is judged in terms of the stationary distribution: terminal values of the optimisation algorithms are binned into each region of the landscape. Noise-in-gradient gradient descent and SAM are observed to skew solutions towards flatter minima (as expected). All algorithms exhibit a lot of noise in the six hump camel example. Next, generalisation performance is examined for four real-world problems: CIFAR10, CIFAR100 (image datasets), and GoEmotion, TweetEval (NLP datasets). Mann-Whitney U tests are performed to test whether significant differences can be detected between generalisation and optimization metrics (they cannot). Median accuracies and standard deviations are reported across each dataset with each optimiser. Hypothesis tests are also performed to compare algorithms to each other to see if significant differences in accuracies can be detected (they cannot). Finally, learning curves are displayed, highlighting that SAM requires double the gradient evaluations per step.

**Strengths:**

- Tackles an important problem: appropriately benchmarking algorithms in a systematic way to inform practitioners of real benefits of one algorithm over another (or lack thereof), rather than testing on individual problems.
- Considers generalisation performance rather than simply rate of convergence of the optimiser (there are too many papers that don't do this).
- A neat collection of synthetic examples are considered.
- Some variety in real-world problems provided (i.e. 2 image datasets, 2 NLP datasets).
- Lots of different examinations.

**Weaknesses:**

- "Rigorous" comparison is far too limited in scope; unambitious. What types of problems, not just landscapes, do certain algorithms perform well on? Consider images vs NLP vs reinforcement learning vs SciML, etc. You need many more real-world examples for such a general analysis to be valuable.
- Generally poorly written, with numerous grammatical issues, inconsistent capitalization (e.g. CIFAR vs Cifar).
- Odd structure; why is Section 4 after Section 3, when Section 3 and Section 5 link with each other?
- Unpleasant presentation: algorithm environments with clashing borders, weird line spacing, figures and tables have text that is too small, weirdness with algorithm text that goes over multiple lines, incorrect citing (citet vs citep).
- Algorithms in the comparison are oddly chosen, and do not comprise a sufficient selection of what is used. Where is Adam for example?
- Hypothesis tests are a bad choice here.
- Multiple hypothesis tests are performed without p-value correction, leading to p < 0.05 conclusions about 5% of the time *due to random chance*.

**Questions:**

- Does GD use the normalised gradient vector, or just the gradient vector?
- Do any of the synthetic examples in Figure 1 correspond to parts of a real loss landscape?
- What is the purpose of the p-values in Section 6.2? Is this to show that there are no meaningful differences in performance between optimisers? Unless realisations are appropriately coupled together (i.e. common random numbers), I'm not sure this is a sensible conclusion. There is naturally a lot of variance in the optimisation procedure, but one optimiser may still perform consistently better than another, everything else equal.
- Can you present the stationary distributions in terms of bar charts rather than as a table? It's really difficult to read as is.
- How long is each optimisation algorithm run for?
- Wouldn't correlation tests be a better idea to test for generalisation vs optimisation section? It is obvious that there should be no significant differences here.
- The differentiation between SetA and SetB seems odd to me. Why not just report all the metrics?

---

> ### Author Response · Authors · 2023-11-22
>
> R: "Considers generalisation performance rather than simply rate of convergence of the optimiser (there are too many papers that don't do this)."
>
> A: We thank the reviewer and now single it out as Contribution 4.
>
> R: "A neat collection of synthetic examples are considered."
>
> A: We thank the reviewer for noting this.
>
> R: " "Rigorous" comparison is far too limited in scope; unambitious. [..] You need many more real-world examples [..]."
>
> A: We have added ImageNet, as well as different model architectures.
>
> R: "Generally poorly written, with numerous grammatical issues, inconsistent capitalization (e.g. CIFAR vs Cifar)."
>
> A: We have carried out a thorough editing and fixed elemental mistakes and typos. We take issue with the blanket statement that the paper is “generally poorly written.”
>
> R: "Odd structure; why is Section 4 after Section 3, when Section 3 and Section 5 link with each other?"
>
> A: Section 3 needs to come first as it introduces the benchmarking setup. Section 4 introduces the algorithms  compared under this setup. Section 5 then relates the performance of these algorithms under the benchmarking setup.
>
> R: "Unpleasant presentation: algorithm environments with clashing borders, [..], incorrect citing (citet vs citep)."
>
> A: These were a side-effect of attempting to pack a lot of content. We have now addressed all.
>
> R: "Algorithms in the comparison are oddly chosen [..]. Where is Adam for example?"
>
> A: We have followed the convention. ADAM has several components that affect behavior, such as the separate learning rate over each weight parameter. This makes it difficult to isolate a “baseline” behavior for deeper understanding that is not confounded by potentially inter-related effects of hyperparameters.
>
> R: "[..] Multiple hypothesis tests are performed without p-value correction [..]."
>
> A: Please note that with our hypothesis testing we are not testing different components and so are not taking the “totality” of tests to reject the null hypothesis. We investigate two distinct approaches to account for potential reviewer concerns re parametric vs. non-parametric.
>
> R: "Does GD use the normalised gradient vector, or just the gradient vector?"
>
> A: We utilize GD on the synthetic functions. Normalizing is computationally more expensive. It also opens up questions. For instance, what step size to use? We determine to proceed with the less expensive (and more popular) version of not normalizing the gradient, and letting the gradient norm provide the speed  towards minima.
>
> R: "Do any of the synthetic examples in Figure 1 correspond to parts of a real loss landscape?"
>
> A: Our choices of synthetic functions that have many local minima and so span the range (including “degenerate” examples such as Beale and Rastrigin in the Supplementary Material) in capturing different characteristics of real landscapes.
>
> R: "What is the purpose of the p-values in Section 6.2? [..] There is naturally a lot of variance [..] but one optimiser may still perform consistently better than another [..]"
>
> A: Comparing SetA by Algorithm X to SetA by Algorithm Y addresses the variance when looking at one model, which most optimization-centric papers do.  Casting our comparison between populations of models via hypothesis testing ought to reveal if one algorithm does “consistently better.”
>
> R: "Can you present the stationary distributions in terms of bar charts rather than as a table? It's really difficult to read as is."
>
> A: We were trying not to delegate too many important points to the Supplementary Material. Bar charts did not "pack in" better. We amplified with graphical visualizations in the Supplementary Material (distribution of end-points over contour plots).
>
> R: "How long is each optimisation algorithm run for?"
>
> A: Section 6 - “[..] 300 epochs.”
>
> R: "[..] It is obvious that there should be no significant differences here."
>
> A:  Expecting no significant differences is akin to believing that you can control for better generalization with better optimization. Researchers are skeptical of this. Correlation analysis is fraught with issues. Even if one were to go down that route, reviewers undoubtedly would insist on a more rigorous comparison under statistical significance testing, which we do in this paper.
>
> R: "The differentiation between SetA and SetB seems odd to me. Why not just report all the metrics?"
>
> A: We are reporting all. We believe this question is due to insufficient differentiation between SetA and SetB, which Contribution #4 now makes.

---

### Official Review · Reviewer_zLAt · 2023-11-04

**Soundness:** 2 fair
**Presentation:** 2 fair
**Contribution:** 1 poor
**Rating:** 3
**Confidence:** 4

**Summary:**

This paper consider SGD, NoiseInModel-GD/SGD, NoiseInGradient-GD/SGD and SAM (Sharpness Aware Minimization) algorithms in the experiments and varying over BH, MonotononicBH, and MetropolisBH.
The authors claimed that they propose a population-based approach to benchmark the algorithms and to better understand the relationship between optimization and generalization.
They argued that to characterize for the behavior of an optimizer one needs a nonlocal view that extends over several trajectories and goes beyond the ”converged”/lowest-loss model.
Thus, they conducted experiment on several trajectories of the algorithms for three synthetic problems and then real world problems.

**Strengths:**

This paper experiments with SGD algorithm and its noise-enabled variants under the Basin Hopping framework. They propose a new procedure for benchmarking the performance of the algorithms: by considering the trajectories created by the algorithms (called populations of models) and comparing their statistical properties. They argue that these trajectories have low loss function and by comparing "populations of models", they can fairly compare two optimizers and avoid conclusions based on one arbitrary or hand-selected model by any optimizer.

**Weaknesses:**

The weaknesses are:
- While collecting additional information from the trajectories of the algorithm is a helpful way to assess the performance better, it also leads to computational cost. The authors should compare this approach to the standard procedure (repeating the experiments multiple times) and take into account the cost into your comparisons.
- The approach is naive in the meaning that they do not consider other factors that may affect the performance of each method: step sizes and other hyper parameters, starting point, whether the methods converge and with what network architecture (for the complex learning problems). Without these considerations, it is difficult to say that the proposed benchmark procedure is more fair than the others.
- Contrary to the contribution sections, there is no 'novel' stochastic optimization algorithm introduced in the paper. The authors only apply previous method to Basin Hopping frameworks.
- The presentation of this paper is poor and very confusing. For examples: "populations of models" actually meant a collection of end-models of each trajectory created by an optimizer. "stationary distribution of an optimizer" makes no sense.

**Questions:**

The authors said they "expand their understanding of the relationship between (loss) optimization, generalization (testing error)". However this point is not clearly explained in the submission. Could you elaborate?

---

> ### Author Response · Authors · 2023-11-22
>
> R: "[..] leads to computational cost. The authors should compare this approach to the standard procedure (repeating the experiments multiple times) and take into account the cost into your comparisons."
>
> A: Copying from Section 6: “We set Tr = 5 and L = 10.” The Tr = 5 trajectories address the reviewer’s suggestion of “repeating the experiments multiple times.” The point on multiple trajectories and computational cost is correct but misses the point. We are not changing the behavior of an algorithm. We are not unfairly making one more expensive than the other.
>
> R: "The approach is naive in the meaning that they do not consider other factors that may affect the performance of each method: step sizes and other hyper parameters, starting point, whether the methods converge and with what network architecture (for the complex learning problems). Without these considerations, it is difficult to say that the proposed benchmark procedure is more fair than the others."
>
> A: The starting point is addressed by varying over trajectories. The point on step sizes and other hyperparameters were explicitly acknowledged in Section 7 (Limitations and Future Work). When researchers debut an optimizer, they “freeze” hyperparameters not introduced in the optimizer. Take SAM; it states several times “see appendix for all hyperparameters.” In the appendix it states: “all other model hyperparameter values are identical to those used in prior work.” Typically, you want to isolate performance in response to a particular effect to start to obtain some understanding. We would love for the community to expand into other factors. Here we focus on better understanding performance by expanding the context first to a population of models, which ought to remove some biases, such as initial conditions.
>
> Our analysis now includes model architectures (summary findings in the main paper and detailed tables in the Supplementary Material). All our findings on the inability to reject the null hypothesis still stand.
>
> We relinquish the point on “fair” and replace it with “new” or “proper” in the title and the rest of the paper.
>
> R: "Contrary to the contribution sections, there is no 'novel' stochastic optimization algorithm introduced in the paper. The authors only apply previous method to Basin Hopping frameworks."
>
> A: We debut novel optimizers that leverage the BH framework. To the best of our knowledge, we cannot find optimizers that do so in published literature on optimization for deep learning.
>
> R: "The presentation of this paper is poor and very confusing. For examples: "populations of models" actually meant a collection of end-models of each trajectory created by an optimizer. "stationary distribution of an optimizer" makes no sense."
>
> A: With this summary term we were trying to incapsulate two distinct analyses, one that compares SetA to SetB for each optimizer for each task, and another that compares SetA of one optimizer to SetA of another optimizer. Hence our "populations of models." We agree  that this may have been confusing and have revised the list of Contributions. In particular, Contribution 4 now explicitly brings to the surface the comparison of what we later refer to as SetA and SetB.
>
> On the point on stationary distributions: You can look at a trajectory as a Markov chain. When converged and over multiple trajectories, you have reached the “stationary distribution.” If the reviewer does not have much appreciation for it, we can remove mention of it. It does not affect any of the analysis or the findings.
>
> R: The authors said they "expand their understanding of the relationship between (loss) optimization, generalization (testing error)". However this point is not clearly explained in the submission. Could you elaborate?
>
> A: With Contribution 4 we aim to address the point raised by the reviewer. Also prompted by another reviewer’s note that the focus on generalization over rate of convergence is good, we now have added the following: “4. Generalization performance over rate of convergence: Unlike most literature on optimization for deep learning, we consider generalization performance rather than simply rate of convergence. We do so over a population of models obtained by an optimizer over several optimization trajectories rather than a single model often obtained as representative of the performance of an optimizer. We compare such a population for its generalization performance (to what we refer as SetA later on in the paper) to a population of models that are sampled by the optimizer and that an oracle has determined have the best generalization performance (to what we refer as SetB later on in the paper). Through this setup we test whether optimization performance is a good proxy of generalization performance utilizing hypothesis testing over populations of models.”

---

### Author Response · Authors · 2023-11-22

We thank the reviewers for taking the time to provide us with feedback. We have addressed every point raised in our responses below, as well as revisions in the paper and Supplementary Material.